# The human cytomegalovirus-encoded pUS28 antagonizes CD4+ T cell recognition by targeting CIITA

**Fabienne Maassen[1,2], Vu Thuy Khanh Le-Trilling[1,3], Luisa Betke[2], Thilo Bracht[4,5], Corinna Siegmund[1], Malte Bayer[4,5], Benjamin Katschinski[1], Antonia Belter[1], Tanja Becker[1†], Denise Mennerich[1], Sebastian Voigt[1], Lori Frappier[6], Barbara Sitek[4,5], Katharina Fleischhauer[2,7], Mirko Trilling[1,3]\***

[1]Institute for Virology, University Hospital Essen, University of Duisburg-Essen, Essen, Germany; [2]Institute for Experimental Cellular Therapy, University Hospital Essen, University of Duisburg-Essen, Essen, Germany; [3]Institute for the Research on HIV and AIDS-associated Diseases, University Hospital Essen, University of Duisburg-Essen, Essen, Germany; [4]Medizinisches Proteom Center, Ruhr University Bochum, Bochum, Germany; [5]Department of Anesthesia, Intensive Care Medicine and Pain Therapy, University Hospital Knappschaftskrankenhaus Bochum, Bochum, Germany; [6]Department of Molecular Genetics, University of Toronto, Toronto, Canada; [7]German Cancer Consortium (DKTK), partner site Essen/Düsseldorf, Heidelberg, Germany

**\*For correspondence:**
mirko.trilling@uni-due.de

**Present address:** [†]Research Center for Emerging Infections and Zoonoses, University of Veterinary Medicine Hannover, Hannover, Germany

**Competing interest:** The authors declare that no competing interests exist.

**Abstract** Human cytomegalovirus (HCMV) is a relevant pathogen, especially for individuals with impaired immunity. Harnessing potent immune antagonists, HCMV circumvents sterile immunity. Given that HCMV prevents the upregulation of *human leukocyte antigen* (HLA)-DP and HLA-DR, we screened a library of HCMV genes by co-expression with the HLA class II (HLA-II)-inducing transcription coordinator *class II transactivator* (CIITA). We identified the latency regulator pUS28 as an interaction factor and potent viral antagonist of CIITA-driven expression of CD74, HLA-DR, HLA-DM, HLA-DQ, and HLA-DP. Both wt-pUS28 and a mutant incapable of inducing G protein-coupled signaling (R129A), but not a mutant lacking the C-terminus, drastically reduced the CIITA protein abundance post-transcriptionally. While control CD4 + T cells from HCMV-seropositive individuals vigorously responded to CIITA-expressing cells decorated with HCMV antigens, pUS28 expression was sufficient to inhibit HLA-II induction and immune recognition by HCMV-specific CD4 + T cells. Our data uncover pUS28 to be employed by HCMV to evade HLA-II-mediated recognition by CD4 + T cells.

## Editor's evaluation

This fundamental study substantially advances our understanding of viral immune evasion by identifying a novel player in this process and uncovering its mode of action. The evidence supporting the conclusions is compelling, with rigorous immunological assays and state-of-the-art biochemistry. The work will be of broad interest to virologists and immunologists.

## Introduction

More than half of the entire adult human population and over 90% of the elderly in developing countries are latently infected with the human cytomegalovirus (HCMV, human betaherpesvirus 5 [HHV-5]; NCBI taxonomy ID 10359) (*Fowler et al., 2022*). In healthy adults, HCMV raises strong canonical as

well as non-canonical immune responses that limit viral replication and, in the vast majority of cases, prevent overt clinical manifestations, while leaving a substantial footprint on the immune system. Accordingly, more than half of all immune parameters are altered in HCMV-infected individuals (*Brodin et al., 2015*). Equipped with an astonishing arsenal of highly potent immune evasins, HCMV circumvents sterile immunity and instead establishes a life-long latency from which it reactivates during episodes of stress or impaired immunity. In cases of immune immaturity, immune senescence, or impairment of the immune system, HCMV replication is not properly controlled, frequently resulting in severe morbidity and mortality in patients such as congenitally infected infants, people living with HIV, and transplant recipients (*Griffiths and Reeves, 2021*).

Although the concerted activity of various mediators of innate and adaptive immunity is crucial for efficient cytomegalovirus (CMV) control (*Polić et al., 1998*), two aspects are particularly important: interferon (IFN) responses and CD4 + T cells. Accordingly, the depletion of CD4 + cells together with IFNγ results in CMV reactivation in more than 75% of animals (*Polić et al., 1998*). IFN secretion is among the first responses elicited upon pathogen encounter. IFNs trigger signaling cascades and initiate a specific transcriptional profile that fosters intrinsic immunity, induces direct innate immunity, and orchestrates adaptive immune responses. In this regard, IFNs act by inducing IFN-stimulated genes (ISGs), many of which are known for their antiviral activity, and by downregulating IFN-repressed genes (IRepGs) (*Megger et al., 2017*; *Schoggins et al., 2011*; *Trilling et al., 2013*). In addition to their ability to establish a cell-intrinsic antiviral state, IFNs boost adaptive immune responses. In particular, IFNγ is well known as a key cytokine for antiviral Th1 responses and as potent enhancer of antigen presentation. In accordance with the relevance of IFNs, the absence of IFN-induced signaling renders mice extremely vulnerable to CMV infections (*Gil et al., 2001*; *Le Trilling et al., 2018*; *Strobl et al., 2005*). CMV infection induces strong CD4 + T cell responses. While best known for their cytokine secretion, helping B cells and CD8 + T cells, CD4 + T cells can elicit antiviral activity (*Jonjić et al., 1990*; *Lim et al., 2020*; *Verma et al., 2016*). The latter is associated with the ability to secrete antiviral cytokines such as IFNγ. Furthermore, certain CD4 + T cells can kill virus-infected cells (*Juno et al., 2017*). HCMV-specific CD4 + T cells were shown to produce IFNγ, TNFα, and granzyme B (*Gamadia et al., 2004*) and to lyse HCMV antigen-expressing cells in vitro (*van Leeuwen et al., 2006*; *van Leeuwen et al., 2004*). Accordingly, CD4 + T cell-mediated immunity is crucial for CMV control in mouse models in vivo (*Jonjić et al., 1989*; *Jonjić et al., 1990*; *Walton et al., 2008*), and for controlling maternal viremia and preventing severe CMV-associated fetal disease during primary rhesus CMV infections (*Bialas et al., 2015*). Furthermore, CD4 + T cells are fundamental to prevent HCMV reactivation in immunocompromised patients receiving solid organ allografts (*Lim et al., 2020*) or hematopoietic cell transplantation (HCT), while insufficient CD4 + T cell levels in transplant recipients are associated with recurrent HCMV reactivation, end-organ disease, and an increased likelihood of lethal infections (*Einsele et al., 1993*; *Gabanti et al., 2015*).

For the canonical T cell receptor (TCR)-dependent activation of CD4 + T cells, antigenic peptides must be presented by the HLA-II molecules HLA-DR, HLA-DQ, or HLA-DP. While antigen-presenting cells (APCs) constitutively express HLA-II, various other cell types start to express HLA-II when they are exposed to IFNγ (*Chang and Flavell, 1995*; *Siegrist et al., 1995*; *Steimle et al., 1993*; *Ting and Trowsdale, 2002*). Activated CD4 + T cells efficiently produce IFNγ, leading to a positive feed-forward loop of increased HLA-II presentation and enhanced CD4 + T cell recognition in the infected niche. Constitutive as well as IFNγ-induced HLA-II expression are both mediated by the class II transactivator (CIITA), which is the essential and sufficient master regulator of the transcription of the genes in the HLA-II locus. Accordingly, CIITA and its co-factors control the expression of the classical HLA-II molecules DR, DQ, DP, the non-classical HLA-II peptidome editors DM and DO, and the invariant chain (Ii, also known as CD74) (*Boss and Jensen, 2003*; *Ting and Trowsdale, 2002*). The absence of functional CIITA results in a loss of HLA-II presentation, leading to a hereditary immunodeficiency called type II bare lymphocyte syndrome (BLS) that causes an extreme vulnerability to infections. In accordance with the relevance of CD4 + T cells and CIITA-driven HLA-II presentation, BLS patients frequently suffer from severe, persistent HCMV infections (*Elhasid and Etzioni, 1996*; *Klein et al., 1993*).

For allogenic hematopoietic cell therapy (HCT), the HLA-DP locus is of particular interest given the frequent incompatibility of patients and donors (*Fleischhauer and Shaw, 2017*), the allotype dependency of peptide presentation and recognition by alloreactive CD4 + T cells (*Arrieta-Bolaños et al., 2022*; *Meurer et al., 2021*; *Yamashita et al., 2017*), and the relevance for chronic virus infections as

indicated by the strong association of HLA-DPB1 SNPs with chronic hepatitis B virus (HBV) infections (see e.g. *Kamatani et al., 2009*). CD4 + T cells recognizing HLA-DP-restricted peptides have also been described for HCMV (*Ameres et al., 2015*; *Hyun et al., 2020*; *Klobuch et al., 2022*; *Stevanovic et al., 2013*; *Ventura et al., 2012*), but to our knowledge, an HCMV-encoded protein antagonizing HLA-DP-restricted antigen presentation has not yet been described.

HCMV-propagating cells such as human fibroblasts, endothelial cells, and epithelial cells are capable of inducing HLA-II presentation after exposure to IFNγ. Moreover, cells serving as a reservoir for HCMV latency, e.g., monocytes, constitutively express HLA-II. CD4 + T cells can recognize HCMV-derived antigens known to be expressed during latency, leading to the recognition of latently infected cells and the production of IFNγ and/or cytotoxic responses (*Mason et al., 2013*). These facts raise the question of how HCMV circumvents CD4 + T cell-mediated elimination during productive replication as well as latency.

Here, we show that the HCMV-encoded G protein-coupled receptor (GPCR) pUS28 directly targets CIITA, impairing HLA-II presentation and CD4 + T cell recognition. Importantly, pUS28 is among the few viral proteins abundantly expressed during both productive replication as well as experimental and natural latency (*Beisser et al., 2001*; *Cheung et al., 2006*; *Goodrum et al., 2002*; *Krishna et al., 2018*; *Krishna et al., 2017a*). Thus, our data reveal a novel mechanism employed by HCMV to circumvent the recognition by CD4 + T cells during different stages of infection.

## Results

### HCMV counteracts IFNγ-induced HLA-DP induction

Given the aforementioned relevance of HLA-DP in HCT and chronic virus infections, we interrogated if and how HCMV affects HLA-DP induction and presentation. IFNγ induced the upregulation of HLA-DP on the surface of HCMV-permissive fibroblasts (*Figure 1A and B*), whereas this was not the case after treatment with IFNα or TNFα (*Figure 1B*). Longer periods of IFNγ incubation led to an increase of HLA-DP and HLA-DR on the cell surface (*Figure 1C*). Despite the fact that the IFNγ-induced induction of HLA-DR and HLA-DP were clearly evident, professional antigen-presenting cells such as monocyte-derived dendritic cells still reached higher HLA-II surface levels (*Figure 1—figure supplement 1*). In clear contrast to uninfected fibroblasts, the upregulation of HLA-DP by IFNγ was prevented in HCMV-infected cells (*Figure 1D and E*). Interestingly, the same inhibitory phenotype was observed upon infection with HCMV mutants lacking the *US2-6* or the *US2-11* gene region (*Figure 1D and E*), that comprise the genes for pUS2 and pUS3, which degrade HLA-DRα and HLA-DMα (*Tomazin et al., 1999*) and block the assembly of HLA-DRα/β heterodimers (*Hegde et al., 2002*), respectively. These data indicate that HCMV encodes so far unknown antagonists of IFNγ-induced HLA-DP presentation located outside of the *US2-11* gene region.

### The HCMV-encoded pUS28 antagonizes CIITA-induced HLA-II expression

Since HCMV prevented HLA-DP induction, we aimed to identify the responsible gene product(s). To this end, we applied an expression library, which has been previously described (*Salsman et al., 2008*), comprising more than 150 canonical HCMV genes, to screen for HLA-DP antagonists. To minimize potentially confounding effects of viral antagonists of IFN signaling acting upstream of CIITA induction, we set up the screen by co-transfecting a CIITA expression plasmid together with individual vectors encoding HCMV proteins. Afterwards, the CIITA-induced HLA-DP cell surface expression was quantified by flow cytometry (*Figure 2A*). The screen provided candidates for the observed antagonism of HLA-DP induction, including pUS9, pUS19, pp65-UL83, and pUS28. For pp65-UL83, *Odeberg et al., 2003* have described a lysosomal sequestration and subsequent degradation of HLA-DR (*Odeberg et al., 2003*), and pUS9 targets several host proteins including HLA/MHC-like molecules such as MR1 and MICA*008 (*Seidel et al., 2015*; *Ashley et al., 2023*). Due to its relevance for the HCMV biology, we further assessed the influence of pUS28 on the CIITA-dependent induction of genes in the HLA-II locus. Validation experiments substantiated that pUS28 was reproducibly capable of significantly diminishing CIITA-induced HLA-DP surface expression (*Figure 2B* and data not shown). Similar to its effect on HLA-DP, pUS28 also significantly diminished the cell surface expression of the presenting molecules HLA-DR (*Figure 2C and D*) and HLA-DQ (*Figure 2D*) as well as the

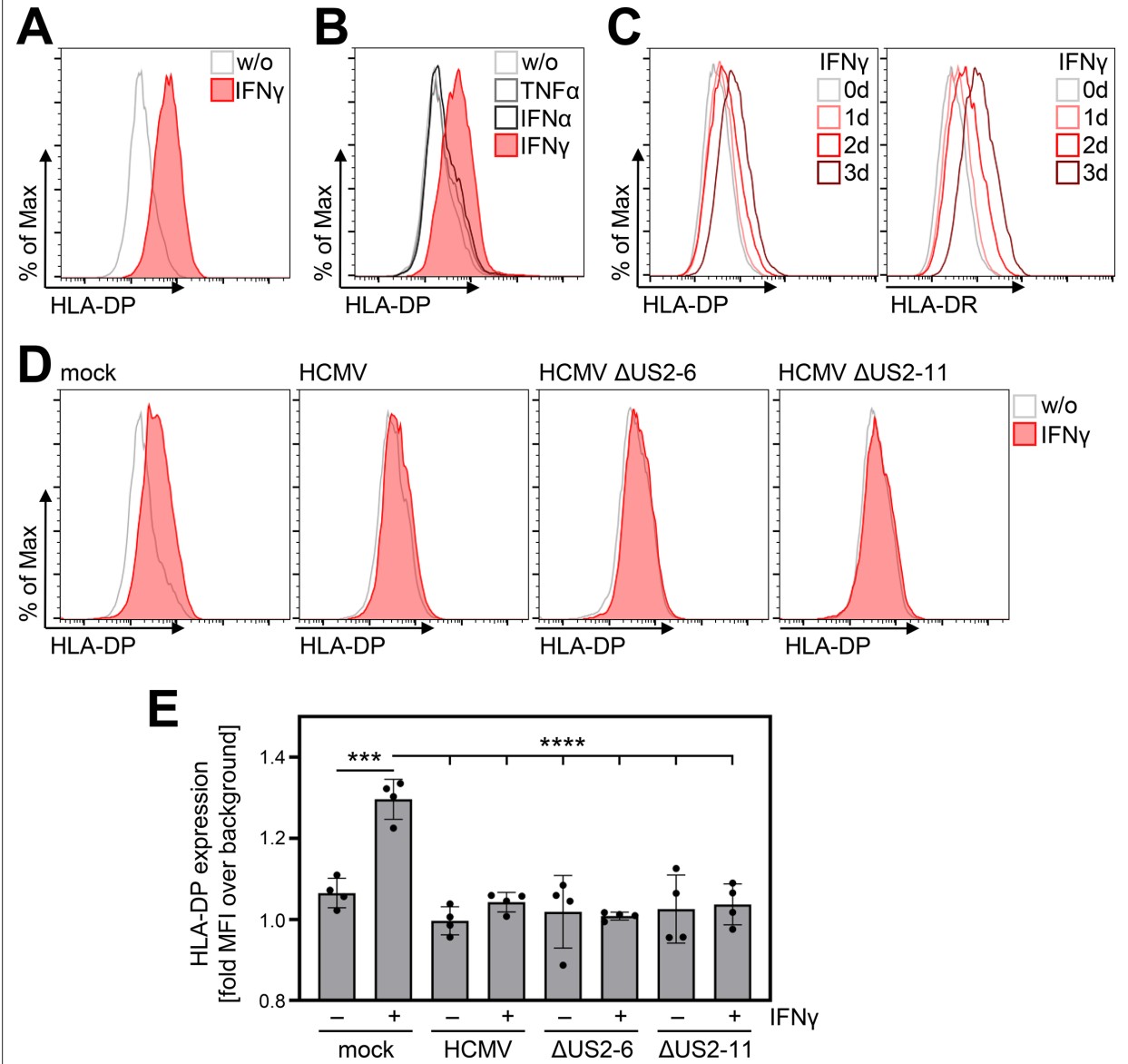

**Figure 1.** IFNγ-induced human leukocyte antigen (HLA)-DP expression is abrogated in Human cytomegalovirus (HCMV)-infected fibroblasts. (**A**) MRC-5 fibroblasts were either left untreated or were treated with 200 U/ml IFNγ. At 72 hr post-treatment, cells were stained with anti-HLA-DP antibody and analyzed by flow cytometry. w/o, untreated. (**B**) MRC-5 fibroblasts were either left untreated or were treated with 200 U/ml IFNγ, 200 U/ml IFNα, or 20 ng/ml TNFα. At 48 hr post-treatment, cells were stained with anti-HLA-DP antibody and analyzed by flow cytometry. w/o, untreated. (**C**) MRC-5 fibroblasts were either left untreated or were treated with 200 U/ml IFNγ for 1, 2, or 3 d. Cells were stained with anti-HLA-DP or anti-HLA-DR antibodies and analyzed by flow cytometry. (**D**) MRC-5 fibroblasts were either mock infected or were infected (MOI 3) with AD169 (HCMV), AD169-BAC2 (HCMVΔUS2-6), or AD169-BAC2ΔUS2-11 (HCMVΔUS2-11). At 4 hr post-infection, cells were treated with 200 U/ml IFNγ. After 48 hr of treatment, cells were stained with anti-HLA-DP antibody and analyzed by flow cytometry. w/o, untreated. (**E**) The mean fluorescence intensity (MFI) values of HLA-DP expression of untreated or IFNγ-treated MRC-5 cells, mock-treated or infected (as in **D**) (n=4). Significance was calculated by two-way ANOVA test. Comparisons are shown when statistically significant.

The online version of this article includes the following figure supplement(s) for figure 1:

**Figure supplement 1.** Human leukocyte antigen (HLA)-DP expression of dendritic cells.

peptide editor HLA-DM (*Figure 2D*). Given that different cell types express varying levels of CIITA, we tested the dose-response relationship between CIITA and pUS28. While increasing amounts of CIITA dose-dependently drove HLA-DP and HLA-DR expression in transfected cells, pUS28 significantly and dose-dependently inhibited surface levels of HLA-DR (*Figure 2E*) and HLA-DP (*Figure 2F*), at different levels of CIITA abundance.

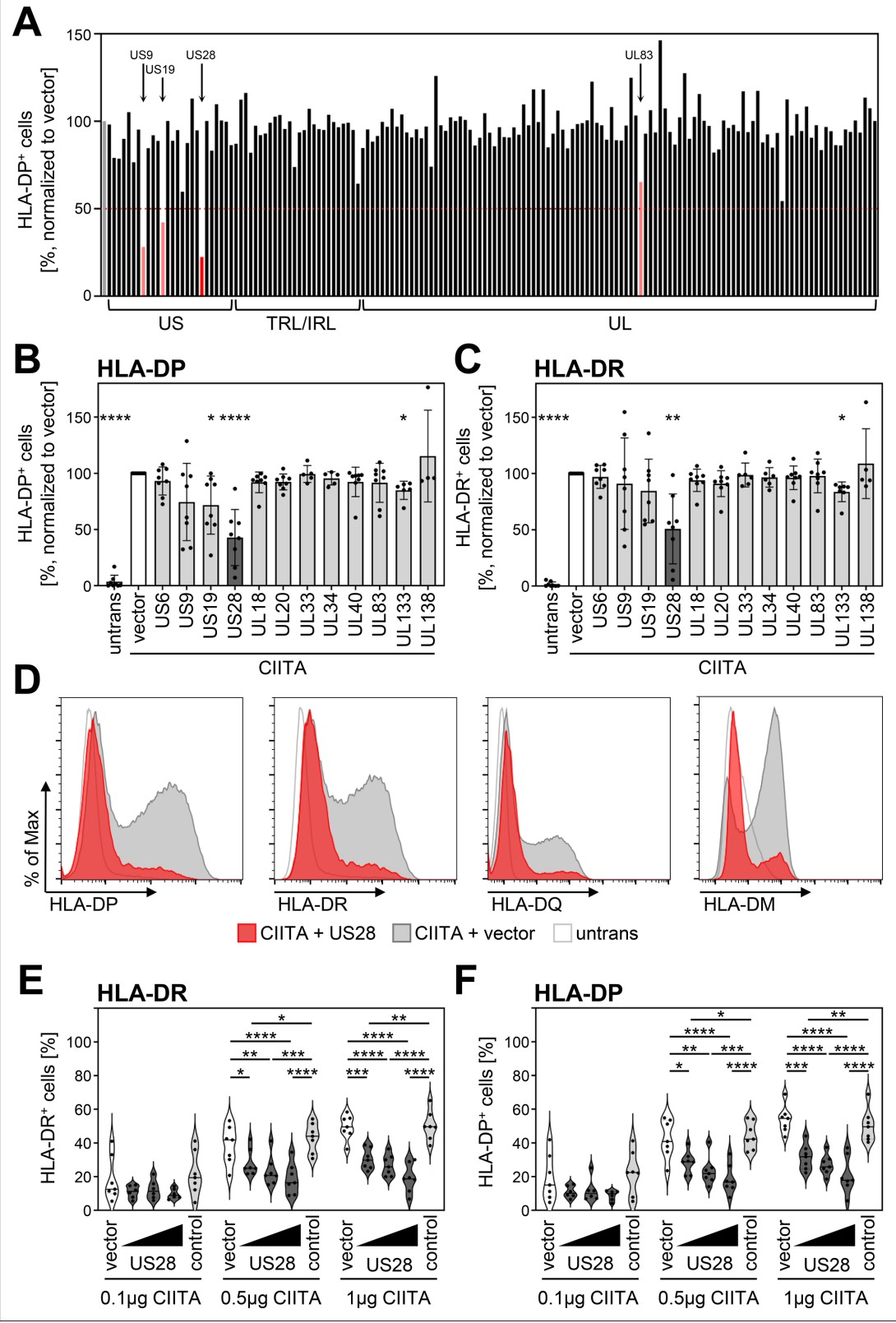

**Figure 2.** Human cytomegalovirus (HCMV)-pUS28 functions as an antagonist of the class II transactivator (CIITA)-induced human leukocyte antigen (HLA) class II upregulation. (**A**) HeLa cells were co-transfected with a CIITA expression construct and a library of single HCMV gene-encoding plasmids or empty vector. At 48 hr post-transfection, cells were stained with anti-HLA-DP antibody and analyzed by flow cytometry. Cell surface expression of HLA-DP was normalized to cells transfected with CIITA expression construct and empty vector. The percentage of HLA-DP-positive cells is shown.

*Figure 2 continued on next page*

*Figure 2 continued*

(**B/C**) HeLa cells were co-transfected with a CIITA expression construct and indicated plasmids or empty vector. At 48 hr post-transfection, cells were stained with anti-HLA-DP (**B**) or anti-HLA-DR (**C**) antibody and analyzed by flow cytometry. Cell surface expression of HLA class II was normalized to cells transfected with CIITA expression construct and empty vector. The percentage of HLA-DP- or HLA-DR-positive cells is shown (mean values ± SD, n=4–8). Significance was calculated by the Kruskal-Wallis test compared to empty vector control. Comparisons are shown when statistically significant. Untrans, untransfected. Vector, empty vector control. (**D**) HeLa cells were either left untreated or were co-transfected with a CIITA expression construct and pcDNA:US28-HA or empty vector. At 48 hr post-transfection, cells were stained with anti-HLA-DP, anti-HLA-DR, or anti-HLA-DQ antibody or intracytoplasmic stained with anti-HLA-DM antibody and analyzed by flow cytometry. (**E/F**) HeLa cells were co-transfected with indicated amounts of CIITA expression construct and increasing doses of pcDNA:US28-HA, a control plasmid (pcDNA:US29-HA), or empty vector. The total DNA amount of each transfection was adjusted to the same level by adding the respective amount of empty vector. At 48 hr post-transfection, cells were stained with anti-HLA-DR (**E**) and anti-HLA-DP (**F**) antibodies and analyzed by flow cytometry. The percentage of HLA-DR- and HLA-DP-positive cells is shown (n=5–8). Significance was calculated by two-way ANOVA test. Comparisons are shown when statistically significant. Vector, empty vector control. Control, control plasmid.

### pUS28 downregulates CIITA post-transcriptionally

To probe into the underlying molecular mechanism of the pUS28-mediated HLA-II inhibition, untagged or epitope-tagged CIITA was co-expressed with pUS28 or a control protein. Afterwards, transcript levels of *CIITA* were quantified by semi-quantitative reverse transcriptase (RT)-PCR. Irrespective of the presence or absence of pUS28, the levels of tagged as well as untagged *CIITA* mRNA remained unaltered (*Figure 3A*), while the amounts of HLA-DP and HLA-DR mRNA were decreased (data not shown). In contrast to the unchanged mRNA levels of *CIITA*, a parallel evaluation of CIITA protein amounts revealed a drastic reduction upon pUS28 co-expression (*Figure 3B*), suggesting a post-transcriptional effect of pUS28 on CIITA. The CIITA down-modulating capacity of pUS28 was observed with different plasmid preparations (*Figure 3C*), and was specific for CIITA, since control proteins such as the enhanced yellow fluorescent protein (EYFP) remained unaffected (*Figure 3C*), arguing against a general influence of pUS28 on transcription or translation. The effect on CIITA was also not an over-arching capacity of viral chemokine receptor homologs, since the protein pUS27, a GPCR encoded by the neighboring gene in the viral genome, did not decrease CIITA protein amounts (*Figure 3D*) or HLA-DP surface levels (*Figure 3E*).

### Global mass spectrometry confirmed the pUS28-mediated decrease of the CIITA abundance and identified downstream targets of the HLA class II pathway

To address the influence of pUS28 on CIITA and the proteome, we performed global mass spectrometry (MS) analyses in which we compared control cells with cells expressing either pUS28, CIITA, or both (*Figure 4A and B*). As expected, our MS analyses confirmed the expression of pUS28 (*Figure 4A-C*). In agreement with previous data, pUS28 led to a significant upregulation of IL-6 (*Slinger et al., 2010*), and attenuated AP-1 components (*Krishna et al., 2019*) (here JunD) (*Figure 4D and E*), validating our experimental setup. These unbiased MS analyses confirmed the significant downregulation of CIITA by pUS28 (*Figure 4F*). Other components of the CIITA enhanceosome, such as RFX5, RFX-AP, NF-YA, and NF-YC, as well as recently described factors (*Kiritsy et al., 2021*) that influence HLA-II transcription showed constitutive, CIITA-independent expression (*Figure 4—figure supplement 1*). Conversely, CD74 - the invariant chain required for proper maturation and loading of HLA class II molecules - was significantly induced by CIITA, unless pUS28 was co-expressed (*Figure 4G*). The negative effect of pUS28 on CIITA-driven CD74 expression was confirmed by flow cytometry (*Figure 4H*). In contrast to several other CIITA-regulated genes, CD74 is not encoded by the HLA-II locus on chromosome 6. Thus, the fact that pUS28 prevents CIITA-induced CD74 induction suggests that pUS28 elicits the negative effect on CIITA target genes independent of their genomic localization in the MHC-II locus.

### The ability of pUS28 to decrease HLA class II expression is evolutionarily conserved and evident in loss-of-function as well as gain-of-function experiments

To validate the effect of pUS28 on CIITA in an independent model, we generated fibroblasts expressing the viral GPCRs pUS28 or pUS27 in a doxycycline-inducible manner. While doxycycline treatment in parental and pUS27-expressing fibroblasts had no significant effect on the IFNγ-induced

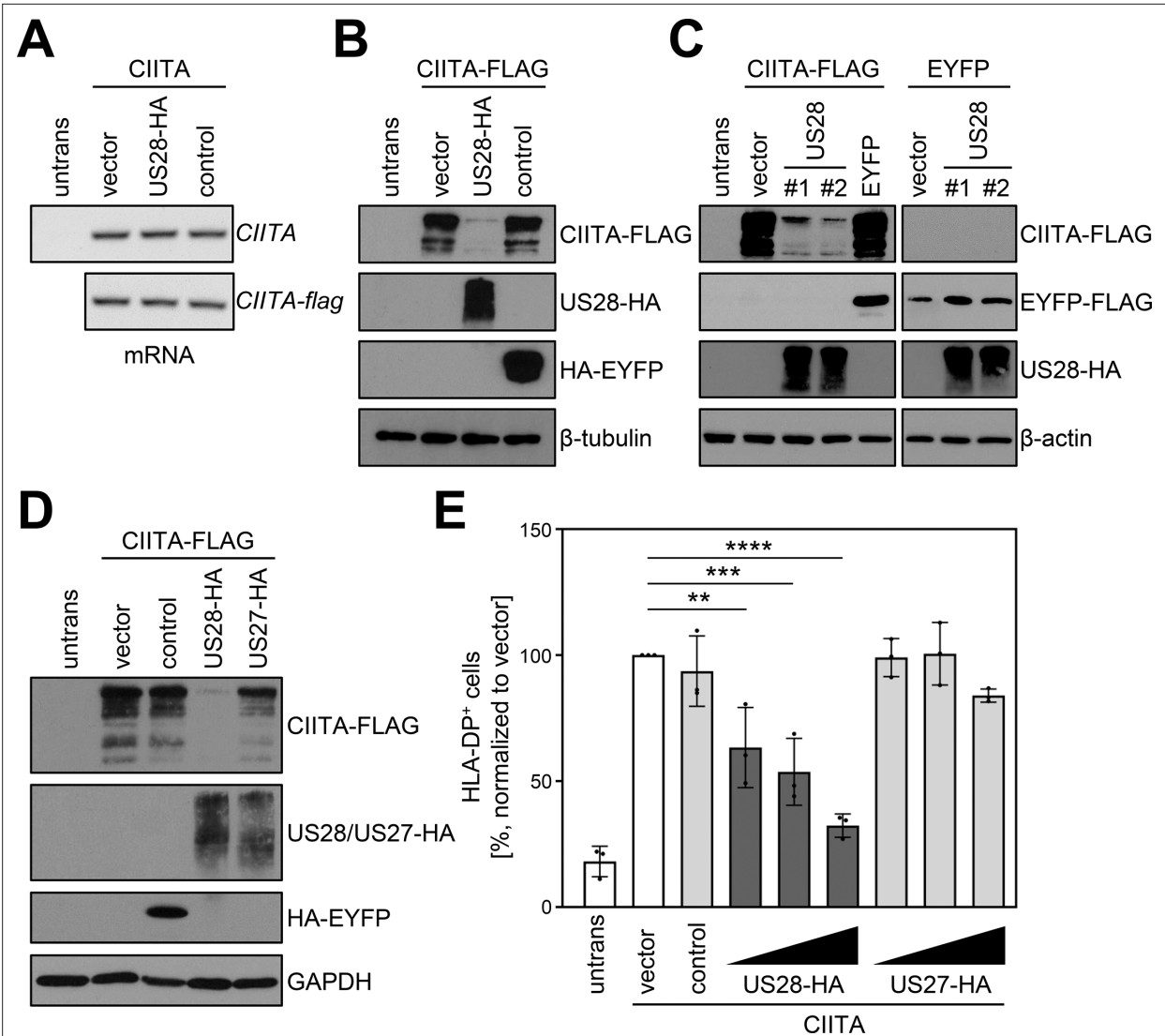

**Figure 3.** pUS28 downregulates class II transactivator (CIITA) post-transcriptionally. (**A/B**) HeLa cells were either left untreated or were co-transfected with CIITA or CIITA-3xFLAG expression constructs and pcDNA:US28-HA, pIRESNeo-FLAG/HA-EYFP (control) or empty vector. At 24 hr post-transfection, cells were harvested and split for preparation of total RNA and protein lysate. RNA samples were used for semi-quantitative RT-PCR with indicated gene-specific primers (**A**) and protein lysates were analyzed by immunoblot using antibodies detecting the indicated proteins or the respective epitope tags (**B**). (**C**) HeLa cells were either left untreated or were co-transfected with the indicated plasmids. At 24 hr post-transfection, protein lysates were generated and analyzed by immunoblot using antibodies detecting the indicated proteins or the respective epitope tags. #1, #2, different plasmid preparations. (**D**) HeLa cells were either left untreated or were co-transfected with CIITA-3xFLAG expression construct and pcDNA:US28-HA, pcDNA:US27-HA, pIRESNeo-FLAG/HA-EYFP (control) or empty vector. At 24 hr post-transfection, protein lysates were generated and analyzed by immunoblot using antibodies detecting the indicated proteins or the respective epitope tags. (**E**) HeLa cells were co-transfected with CIITA expression construct and increasing doses of pcDNA:US28-HA or pcDNA:US27-HA, pIRESNeo-FLAG/HA-EYFP (control) or empty vector. The total DNA amount of each transfection was adjusted to the same level by adding the respective amount of empty vector. At 48 hr post-transfection, cells were stained with anti-human leukocyte antigen (HLA)-DP antibody and analyzed by flow cytometry. The percentage of HLA-DP-positive cells normalized to empty vector control is shown (n=3). The different transfection conditions were compared to the control condition (vector) by one-way ANOVA test. Comparisons are shown when statistically significant. Untrans, untransfected. Vector, empty vector control. Control, pIRESNeo-FLAG/HA-EYFP.

The online version of this article includes the following source data for figure 3:

**Source data 1.** PDF files containing original western blots for *Figure 3A-D*, indicating the relevant bands and treatments.

**Source data 2.** Original files for western blot analysis displayed in *Figure 3A-D*.

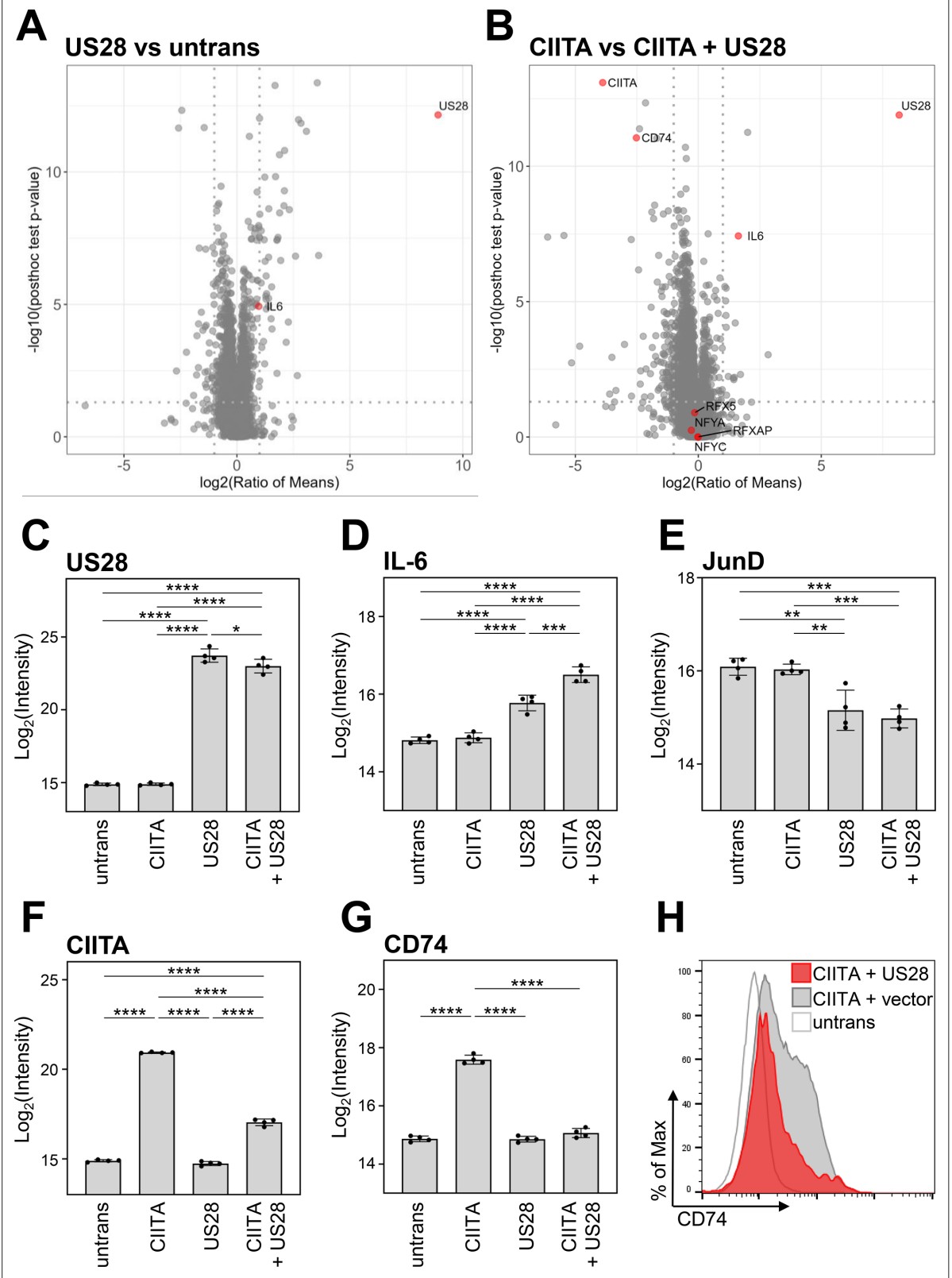

**Figure 4.** Global proteome analysis revealed that pUS28 targets class II transactivator (CIITA) and affects the CIITA-regulated protein CD74. (**A/B**) HeLa cells were either left untreated or were transfected with CIITA-3xFLAG expression construct, pcDNA:US28-HA, or both. At 24 hr post-transfection, whole cell lysates were generated and subjected to mass spectrometric analysis of protein abundance. Volcano plots showing log2 (ratio of means) (x-axis) versus significance (y-axis) of the comparison of untreated and pUS28-expressing cells (**A**) or cells expressing CIITA in the presence or absence of pUS28

*Figure 4 continued on next page*

*Figure 4 continued*

(**B**). Proteins are indicated as gray dots, highlighted in red are IL-6 (known to be upregulated by pUS28), co-factors of HLA-II transcription and signaling, CIITA, and pUS28. (**C–G**) Changes in the abundance of selected proteins detected by MS: (**C**) pUS28, (**D**) IL-6, (**E**) JunD, (**F**) CIITA, (**G**) CD74. Depicted are log2 (intensity) values of untreated cells, cells expressing either CIITA, pUS28, or both (n=4). Significance was calculated by one-way ANOVA test. Comparisons are shown when statistically significant. (**H**) HeLa cells were either left untreated or were co-transfected with a CIITA expression construct and pcDNA:US28-HA or empty vector. At 48 hr post-transfection, cells were stained with anti-CD74 antibody and analyzed by flow cytometry. Untrans, untransfected. Vector, empty vector control.

The online version of this article includes the following figure supplement(s) for figure 4:

**Figure supplement 1.** Global proteome analysis showed that the other components of the CIITA enhanceosome and recently described regulators of HLA-II transcription are constitutively expressed.

HLA-DP expression, HLA-DP cell surface levels were significantly decreased upon pUS28 induction (*Figure 5*). Additionally, an HCMV mutant lacking the gene *US28* (ΔUS28-HCMV) was generated and tested regarding its effect on HLA-II. Leukemic cells that constitutively express HLA-II were either infected with wild-type (wt) HCMV or ΔUS28-HCMV. In accordance with aforementioned data, wt HCMV significantly decreased the percentage of HLA-DP- and HLA-DR-positive cells in comparison to mock-treated cells. In contrast, ΔUS28-HCMV showed an impaired ability to downregulate HLA-II (*Figure 5B and C*). We also assessed the effect of the mouse cytomegalovirus (MCMV)-encoded pUS28 homolog pM33 regarding its capacity to downregulate human (*Figure 5D*) and mouse CIITA (*Figure 5E*). We found that pM33 and pUS28 both downregulate human and mouse CIITA, suggesting an evolutionarily conserved function.

## pUS28 interacts with CIITA and reduces its half-life irrespective of the G protein-coupling capacity

Next, we tested if pUS28 and CIITA physically interact. To circumvent the issue that CIITA levels were diminished to almost undetectable levels upon pUS28 co-expression, lysates either containing pUS28 or CIITA were combined before an immunoprecipitation (IP) was performed. The IP of pUS28 co-purified CIITA (*Figure 6A*, upper panel) and vice versa (*Figure 6A*, lower panel), indicating that both proteins form physical complexes. To test if pUS28 forces CIITA into a detergent-insoluble fraction, cell lysates were prepared with a denaturing lysis buffer (based on high urea concentrations) and subjected to immunoblot analysis. This approach did not lead to the reappearance of CIITA (*Figure 6—figure supplement 1*). Irrespective of the absence or presence of pUS28, we could not detect CIITA in cell culture supernatants (*Figure 6—figure supplement 2*), arguing against pUS28-mediated CIITA shedding. Since we did not find evidence for sequestration or shedding, we tested for a pUS28-mediated CIITA degradation. To this end, we compared the half-life of CIITA in the presence or absence of pUS28. Despite the inherently short half-life of CIITA (*Schnappauf et al., 2003*), we observed a more rapid CIITA decay when pUS28 was co-expressed (*Figure 6B* and *Figure 6—figure supplement 3*).

Intriguingly, a panel of inhibitors targeting different cellular degradation pathways (e.g. the inhibitors of the ubiquitin-proteasome pathway MG-132 and Bortezomib, the inhibitors of UBA3-dependent neddylation blocking cullin-RING ubiquitin ligase activity MLN4924 and TAS4464, the autophagy inhibitors bafilomycin and 3-MA, the inhibitors of lysosomal acidification chloroquine and ammonium chloride, the pan-caspase inhibitor Z-VAD-FMK, the protease inhibitors Pepstatin A, E-64, PMSF, and a pan-protease inhibitor cocktail, as well as the dynamin inhibitor Dynasore and the convertase inhibitor Decanoyl-RVKR-CMK) all failed to significantly and reproducibly restore CIITA levels in pUS28-expressing cells (*Figure 6—figure supplement 4* and data not shown), implying that pUS28 reduces CIITA levels either by redundant or unusual degradation processes.

To assess if the G-protein signaling of pUS28 is essential for the downregulation of CIITA, we compared wt-pUS28 and R129A-pUS28. The latter is a well-studied mutant of pUS28 that harbors a single amino acid substitution at position 129 (arginine to alanine) within the canonical DRY motif, abrogating G-protein signaling without compromising the subcellular localization or internalization (*Waldhoer et al., 2003*; visualized in *Figure 7A*). Both pUS28 variants diminished CIITA protein levels (*Figure 7B*). Conversely, the intracellular C terminus turned out to be indispensable for the CIITA degradation (*Figure 7C*). In accordance with these findings, R129A-pUS28 and wt-pUS28 similarly diminished the surface expression of HLA-DP (*Figure 7D*) and HLA-DR (*Figure 7E*), while the mutant

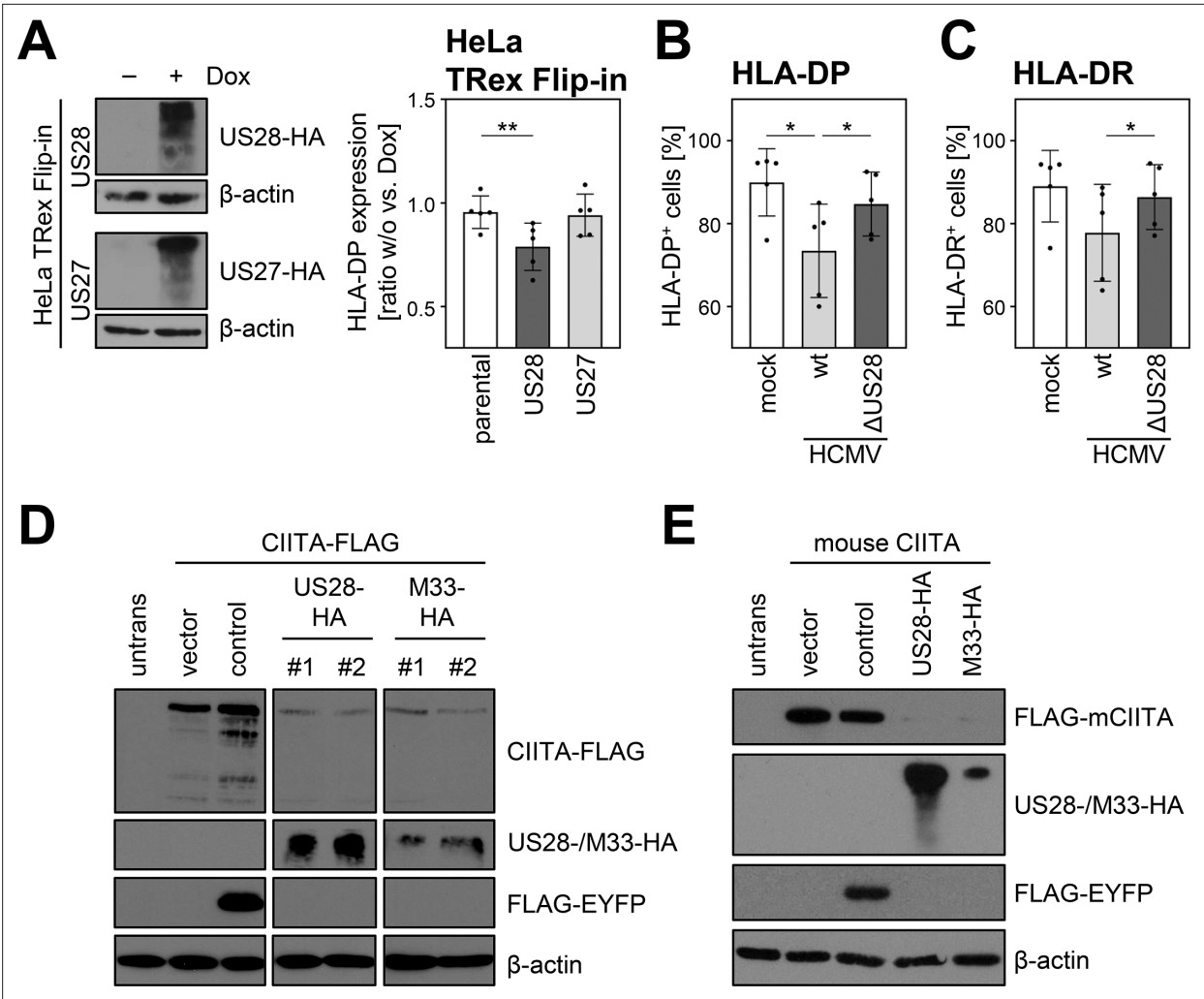

**Figure 5.** The ability of pUS28 to decrease human leukocyte antigen (HLA) class II expression is evolutionary conserved and evident in loss-of-function as well as gain-of-function experiments. (**A**) HeLa TRex Flip-in:US28HA (US28) and HeLa TRex Flip-in:US27HA (US27) cells were either treated with 200 µg/ml doxycycline or left untreated. At 24 hr after treatment, protein lysates were generated and analyzed by immunoblot using antibodies detecting the indicated proteins or the respective epitope tags. Furthermore, HeLa TRex Flip-in (parental), HeLa TRex Flip-in:US28HA (US28), and HeLa TRex Flip-in:US27HA (US27) were either treated with 200 U/ml IFNγ or a combination of 200 µg/ml doxycycline and 200 U/ml IFNγ. After 48 hr of treatment, cells were stained with anti-HLA-DP antibody and analyzed by flow cytometry. The fold induction of the mean fluorescence intensity (MFI) values of HLA-DP expression of cells in presence compared to absence of doxycycline treatment (taking the change over background into account) are shown (n=5). Significance was calculated by one-way ANOVA test. Comparisons are shown when statistically significant. (**B/C**) UoC-B6 cells were either mock infected or were infected (MOI 3) with AD169-BAC2-UL131rep (wt) or AD169-BAC2-UL131repΔUS28 (ΔUS28). At 2 hr post-infection, medium change was performed. At 3 d post-infection, cells were stained with anti-HLA-DP and anti-HLA-DR antibodies and analyzed by flow cytometry. The percentage of HLA-DP- (**B**) and HLA-DR-positive (**C**) cells is shown (n=5). Significance was calculated by RM one-way ANOVA test. Comparisons are shown when statistically significant. (**D**) HeLa cells were either left untreated or were co-transfected with CIITA-3xFLAG expression construct and pcDNA:US28-HA, pcDNA:M33-HA, pIRESNeo-FLAG/HA-EYFP (control) or empty vector pcDNA3.1. At 24 hr post-transfection, protein lysates were generated and analyzed by immunoblot using antibodies detecting the indicated proteins or the respective epitope tags. #1, #2, different plasmid preparations. (**E**) HeLa cells were either left untreated or were co-transfected with 3xFLAG-mouseCIITA expression construct and pcDNA:US28-HA, pcDNA:M33-HA, pIRESNeo-FLAG/HA-EYFP (control) or empty vector. At 24 hr post-transfection, protein lysates were prepared and analyzed by immunoblot using antibodies detecting the indicated proteins or the respective epitope tags.

The online version of this article includes the following source data for figure 5:

**Source data 1.** PDF files containing original western blots for ***Figure 5A, D and E***, indicating the relevant bands and treatments.

**Source data 2.** Original files for western blot analysis displayed in ***Figure 5A, D and E***.

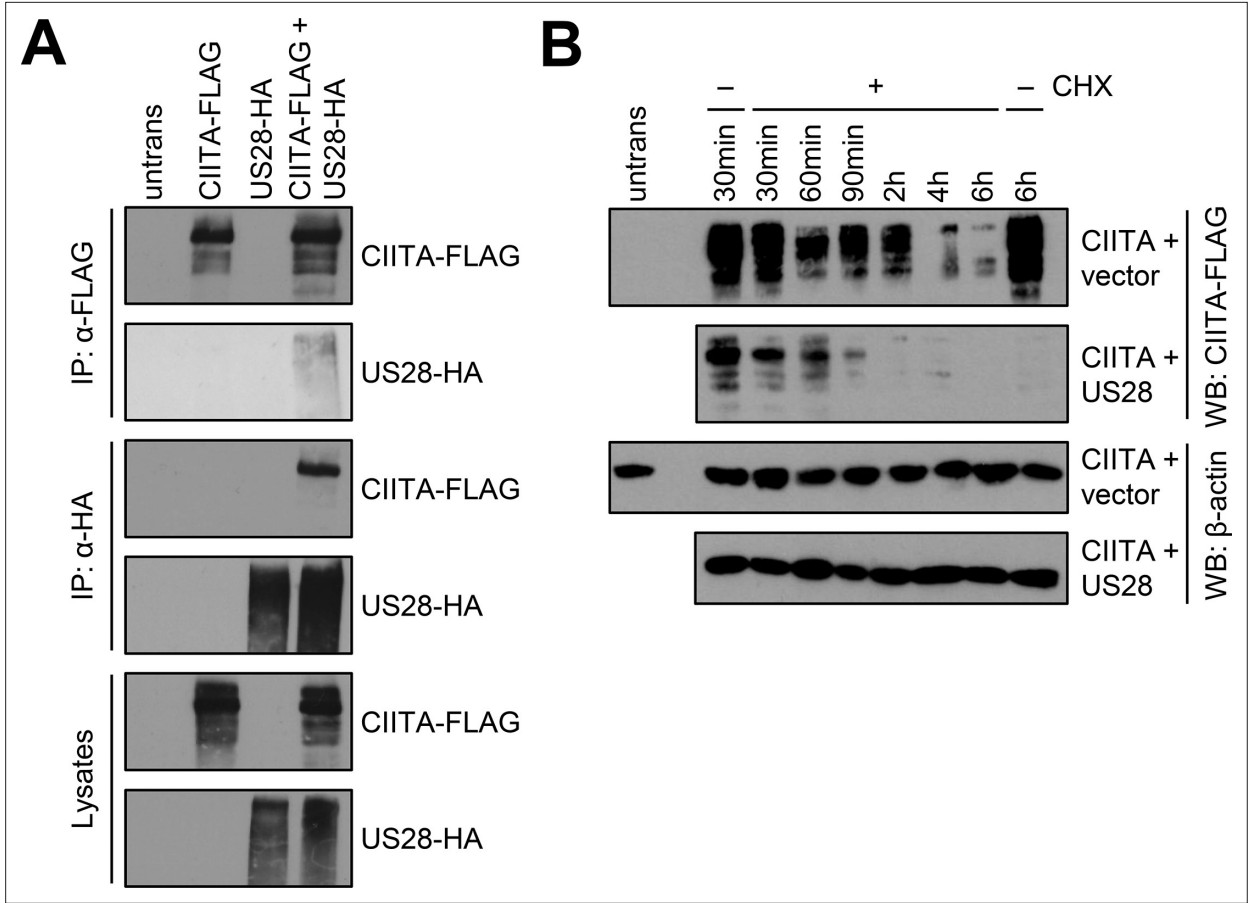

**Figure 6.** pUS28 physically interacts with class II transactivator (CIITA) and reduces its half-life. (**A**) HeLa cells were either left untreated or were transfected with CIITA-3xFLAG expression construct or pcDNA:US28-HA. At 24 hr post-transfection, protein lysates were generated and an IP with HA- or FLAG-specific mouse monoclonal antibodies was performed either with unmodified samples or with mixed samples of CIITA- and US28-transfected cells. The lysates and IP samples were analyzed by immunoblot test CIITA and pUS28 co-precipitation. (**B**) HeLa cells were co-transfected with CIITA-3xFLAG expression construct and pcDNA:US28-HA or empty vector. At 16 hr post-transfection, cells either were left untreated or were incubated with 50 µg/ml cycloheximide (CHX) for indicated periods. Protein lysates were generated and analyzed by immunoblot using antibodies detecting the indicated proteins or the respective epitope tags. All samples were run on one gel and detected on the same membrane (*Figure 5*). For better comparison, US28 samples were presented underneath the empty vector samples.

The online version of this article includes the following source data and figure supplement(s) for figure 6:

**Source data 1.** PDF files containing original western blots for *Figure 6A-B*, indicating the relevant bands and treatments.

**Source data 2.** Original files for western blot analysis displayed in *Figure 6A-B*.

**Figure supplement 1.** pUS28 diminishes class II transactivator (CIITA) protein levels also in denaturing lysis buffer.

**Figure supplement 1—source data 1.** PDF files containing original western blots for *Figure 6—figure supplement 1*, indicating the relevant bands and treatment.

**Figure supplement 1—source data 2.** Original files for western blot analysis displayed in *Figure 6—figure supplement 1*.

**Figure supplement 2.** The loss of class II transactivator (CIITA) in pUS28-expressing cells is not caused by CIITA shedding as indicated by its absence in cell supernatants.

**Figure supplement 2—source data 1.** PDF files containing original western blots for *Figure 6—figure supplement 2*, indicating the relevant bands and treatments.

**Figure supplement 2—source data 2.** Original files for western blot analysis displayed in *Figure 6—figure supplement 2*.

**Figure supplement 3.** pUS28 reduces the half-life of class II transactivator (CIITA).

**Figure supplement 4.** Inhibition of the usual protein degradation pathways does not restore class II transactivator (CIITA) protein levels in the presence of pUS28.

**Figure supplement 4—source data 1.** PDF files containing original western blots for *Figure 6—figure supplement 4*, indicating the relevant bands and treatments.

*Figure 6 continued on next page*

*Figure 6 continued*

**Figure supplement 4—source data 2.** Original files for western blot analysis displayed in *Figure 6—figure supplement 4*.

lacking the C terminus (US28-ΔC) did not (*Figure 7F and G*), indicating that pUS28 targets CIITA through its C terminus. In IP experiments, pUS28-ΔC co-purified CIITA (*Figure 7H*), suggesting that the lack of the C terminus of pUS28 did not impair the physical interaction with CIITA but rather affected the degradation machinery. Further investigations of the C-terminal part of pUS28 revealed that amino acids in proximity to the transmembrane domain 7 were required for downregulation of CIITA as only a mutant lacking the whole C terminus completely lost the capability to target CIITA (*Figure 7I*).

## pUS28 antagonizes HCMV-specific CD4+ T cells

After identifying pUS28 as a viral antagonist of the CIITA-driven HLA class II expression and considering that pUS28 is expressed during HCMV latency, we aimed to investigate the relevance of pUS28 for immune recognition. HCMV-specific CD4 + T cells were enriched by antigen-dependent expansion using peripheral blood mononuclear cells (PBMCs) from HCMV-seropositive healthy individuals exposed to lysates derived from HCMV-infected fibroblasts (see schematic overview in *Figure 8A*). During the 24 hr re-stimulation phase, the HCMV-specific CD4 + T cells were co-cultured with HCMV antigen-loaded fibroblasts expressing CIITA together with pUS28, with pUS28-R129A or with an irrelevant protein. CD4 + T cell activation was evaluated by CD137 upregulation. HCMV-specific CD4 + T cells from HCMV-positive individuals vigorously responded to CIITA-expressing cells, but not to control cells that did not express CIITA (*Figure 8B* and *Figure 8—figure supplement 1*), indicating a specific HLA-II-dependent T cell activation. In the presence of pUS28 or the mutant R129A-pUS28, the immune recognition of CIITA-expressing, antigen-loaded cells by HCMV-specific CD4 + T cells was significantly reduced (*Figure 8B*). Since CD4 + T cells are well-known for the ability to produce antiviral cytokines such as IFNγ, we assessed the cell culture supernatants collected after the 24 hr re-stimulation phase regarding their antiviral activity against HCMV. We conditioned human fibroblasts with the supernatants prior to the infection with an EGFP-expressing reporter HCMV. Afterwards, we quantified HCMV-induced EGFP expression (*Figure 8C*) and visualized the degree of HCMV infection by microscopy (*Figure 8D*). Supernatants from HCMV-specific CD4 + T cells stimulated with CIITA-expressing cells strongly inhibited HCMV-induced EGFP expression, while supernatants derived from CD4 + T cells stimulated in the presence of pUS28 showed diminished antiviral activity (*Figure 8C and D*). These data demonstrate that pUS28 inhibits the CIITA-driven and HLA-II-dependent CD4 + T cell activation in terms of the production of antiviral cytokines.

## Discussion

In this study, we identified pUS28 as HCMV-encoded antagonist of CIITA and CIITA-driven HLA class II expression (*Figure 9*). The viral protein was found to physically interact with CIITA, causing a post-transcriptional decline of the CIITA protein by an increased protein decay. This CIITA degradation was sufficient to decrease HLA-DR, HLA-DQ, HLA-DM, CD74/invariant chain, and HLA-DP expression and to abrogate activation of HCMV-specific CD4 + T cells. Despite its role as a highly relevant human pathogen, attenuated HCMVs became the basis for the development of promising vaccine vectors (e.g. for the vaccination against lentiviruses *Picker et al., 2023*), which may benefit from the deletion of immune evasins such as US28, either in terms of enhanced immunogenicity and/or increased safety.

Briefly after its discovery as HCMV-encoded GPCR (*Chee et al., 1990*), the striking homology of pUS28 to human CX3CR1 was noticed (*Gao et al., 1993*) and its functionality as calcium-mobilizing beta chemokine receptor was documented (*Gao and Murphy, 1994*). Since then, pUS28 has became one of the best-studied HCMV proteins, and several important immunological and viral functions have been assigned to this molecule. For example, it can act as a co-factor for the HIV entry (*Pleskoff et al., 1997*), it mediates G protein-coupled intracellular signaling (*Billstrom et al., 1998*), it sequesters RANTES/MCP-1 (*Bodaghi et al., 1998*) and serves as fractalkine receptor (*Burg et al., 2015*; *Kledal et al., 1998*), it mediates vascular smooth muscle cell migration (*Burg et al., 2015*; *Kledal et al., 1998*; *Melnychuk et al., 2004*; *Streblow et al., 1999*; *Streblow et al., 2003*), and it may act as an oncogene or onco-modulatory protein (*Bongers et al., 2010*; *Maussang et al., 2006*; *Slinger*

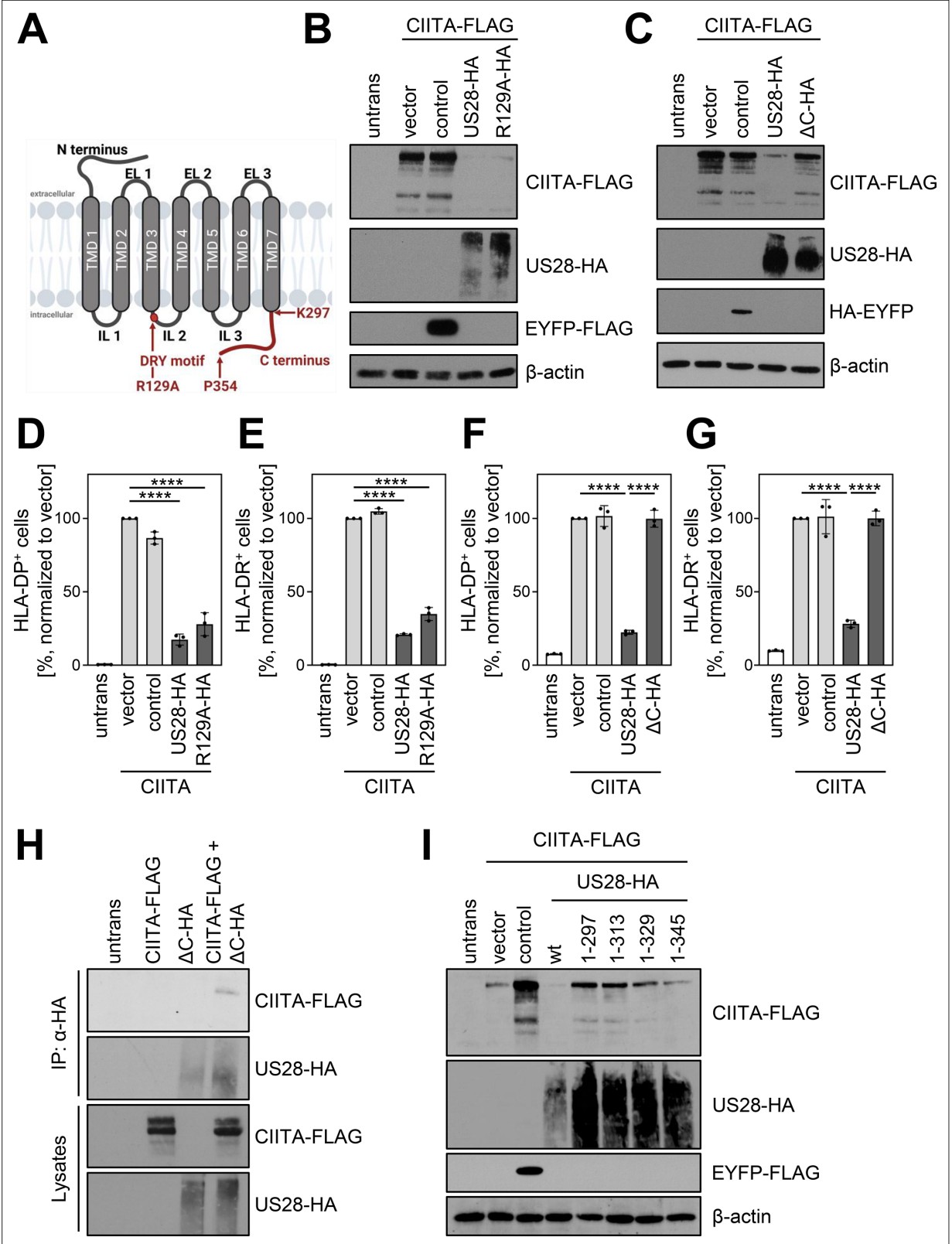

**Figure 7.** pUS28 targeting class II transactivator (CIITA) is independent of the G-protein coupling capacity but requires the C terminus of pUS28. (**A**) Schematic overview of the structure of pUS28. All structural parts of the protein and the DRY motif are indicated. Mutation of the arginine in this motif to alanine (R129A) ablates G-protein coupling. This panel was created using BioRender.com. (**B**) HeLa cells were either left untreated or were co-transfected with CIITA-3xFLAG expression construct and pcDNA:US28-HA, pcDNA:US28-R129A-HA, pIRESNeo-FLAG/HA-EYFP, or empty vector.

*Figure 7 continued on next page*

*Figure 7 continued*

At 24 hr post-transfection, protein lysates were generated and analyzed by immunoblot using antibodies detecting the indicated proteins or the respective epitope tags. (**C**) HeLa cells were either left untreated or were co-transfected with CIITA-3xFLAG expression construct and pcDNA:US28-HA, pcDNA:US28-ΔC-HA, pIRESNeo-FLAG/HA-EYFP (control) or empty vector. At 24 hr post-transfection, protein lysates were generated and analyzed by immunoblot using antibodies detecting the indicated proteins or the respective epitope tags. (**D/E**) HeLa cells were either left untreated or were co-transfected with CIITA expression construct and pcDNA:US28-HA, pcDNA:US28-R129A-HA, pIRESNeo-FLAG/HA-EYFP (control) or empty vector. At 48 hr post-transfection, cells were stained with anti-HLA-DP (**D**) or anti-HLA-DR (**E**) antibodies and analyzed by flow cytometry. Cell surface expression of HLA-DP or HLA-DR was normalized to cells co-transfected with CIITA expression construct and empty vector. The percentage of HLA-DP- and HLA-DR-positive cells is shown (n=3). The different transfection conditions were compared to the control condition (vector) by one-way ANOVA test. Additionally, US28-R129A-HA was compared to US28-HA. Comparisons are shown when statistically significant. (**F/G**) HeLa cells were either left untreated or were co-transfected with CIITA-3xFLAG expression construct and pcDNA:US28-HA, pcDNA:US28-ΔC-HA, pIRESNeo-FLAG/HA-EYFP (control) or empty vector. At 48 hr post-transfection, cells were stained with anti-HLA-DP (**F**) or anti-HLA-DR (**G**) antibodies and analyzed by flow cytometry. Cell surface expression of HLA-DP or HLA-DR was normalized to cells transfected with CIITA expression construct and empty vector. The percentage of HLA-DP- and HLA-DR-positive cells is shown (n=3). The different transfection conditions were compared to the control condition (vector) by one-way ANOVA test. Additionally, US28-ΔC-HA was compared to US28-HA. Comparisons are shown when statistically significant. (**H**) HeLa cells were either left untreated or were transfected with CIITA-3xFLAG expression construct or pcDNA:US28-ΔC-HA. At 24 hr post-transfection, protein lysates were generated and an IP with HA-specific mouse monoclonal antibody was performed either with unmodified samples or with mixed samples of CIITA- and US28-transfected cells. The lysates and IP samples were analyzed by immunoblot to test CIITA and pUS28-ΔC co-precipitation. (**I**) HeLa cells were either left untreated or were co-transfected with CIITA-3xFLAG expression construct and pIRES:US28-HA, different pIRES:US28-ΔC-HA mutants, pIRESNeo-FLAG/HA-EYFP (control) or empty vector. At 24 hr post-transfection, protein lysates were generated and analyzed by immunoblot using antibodies detecting the indicated proteins or the respective epitope tags. Untrans, untransfected. Vector, empty vector control. ΔC (1-297), C-terminal (aa 298–354) deletion mutant of pUS28. 1–313, C-terminal (aa 314–354) deletion mutant of pUS28. 1–329, C-terminal (aa 330–354) deletion mutant of pUS28. 1–345, C-terminal (aa 346–354) deletion mutant of pUS28. R129A, point mutation mutant of pUS28. Control, pIRESNeo-FLAG/HA-EYFP.

The online version of this article includes the following source data for figure 7:

**Source data 1.** PDF files containing original western blots for *Figure 7A-B and H-I*, indicating the relevant bands and treatments.

**Source data 2.** Original files for western blot analysis displayed in *Figure 7A-B and H-I*.

*et al., 2010*). In terms of its expression profile, pUS28 is special due to its presence in experimental and natural latency (*Beisser et al., 2001*; *Cheung et al., 2006*; *Goodrum et al., 2002*; *Krishna et al., 2018*; *Krishna et al., 2017a*), constituting a drug target for the latent HCMV reservoir using specifically designed toxins (*Krishna et al., 2017b*). Our study adds the inhibition of CIITA-induced HLA-II antigen presentation to the list of important pUS28 functions. The relevance of pUS28 for latency establishment and reactivation has been studied in huNSG mice (*Crawford et al., 2019*). These mice develop certain aspects of a human T cell compartment (*Legrand et al., 2006*), which may lead to HLA-II-restricted elimination of infected cells by CD4 + T cells. In addition to its influence on latency and reactivation, the herein documented ability of pUS28 to counteract CIITA-driven HLA-II presentation may influence the outcome of such experiments by effects of pUS28 on the CD4 + T cell recognition by the immune system. Similarly, pUS28-expressing mouse cytomegalovirus (MCMV) mutants have been studied in mice (*Farrell et al., 2013*; *Legrand et al., 2006*). In this regard, it may be relevant to highlight that the HCMV-encoded pUS28 and its MCMV homolog pM33 are both capable of downregulating mouse CIITA (*Figure 5E*).

Elegant work by the Sinclair laboratory revealed that pUS28, besides its function as GPCR and latency regulator, targets certain host proteins such as MNDA/PYHIN3 and IFI16/PYHIN2 for rapid degradation. In contrast to the effect on CIITA shown here, however, this degradation was dependent on GPCR signaling as indicated by the restoration of target protein levels when R129 was mutated (*Elder et al., 2019*). Although CIITA and HLA-DP levels were neither assessed nor discussed, the same work documented a negative effect of pUS28 on constitutive HLA-DR expression, while no difference between wt-pUS28 and an R129A mutant with regard to the IFNγ-induced HLA-DR expression was observed (*Elder et al., 2019*). Importantly, *Elder et al., 2019* and *Slobedman et al., 2002* showed that latently infected primary human CD14 + monocytes and human fetal liver hematopoietic cells exhibit less HLA-DR on the surface compared to uninfected cells (*Elder et al., 2019*; *Slobedman et al., 2002*). Our data regarding the targeting of CIITA by pUS28 now provide a parsimonious molecular explanation for these findings, and extend it to other highly relevant HLA-II molecules including CD74, HLA-DM, HLA-DQ, and HLA-DR – which is very well in line with findings by *Shnayder et al., 2020* who showed an inverse correlation between HCMV transcripts and CD74 as well as HLA-II expression in CD14 + monocytes (*Shnayder et al., 2020*).

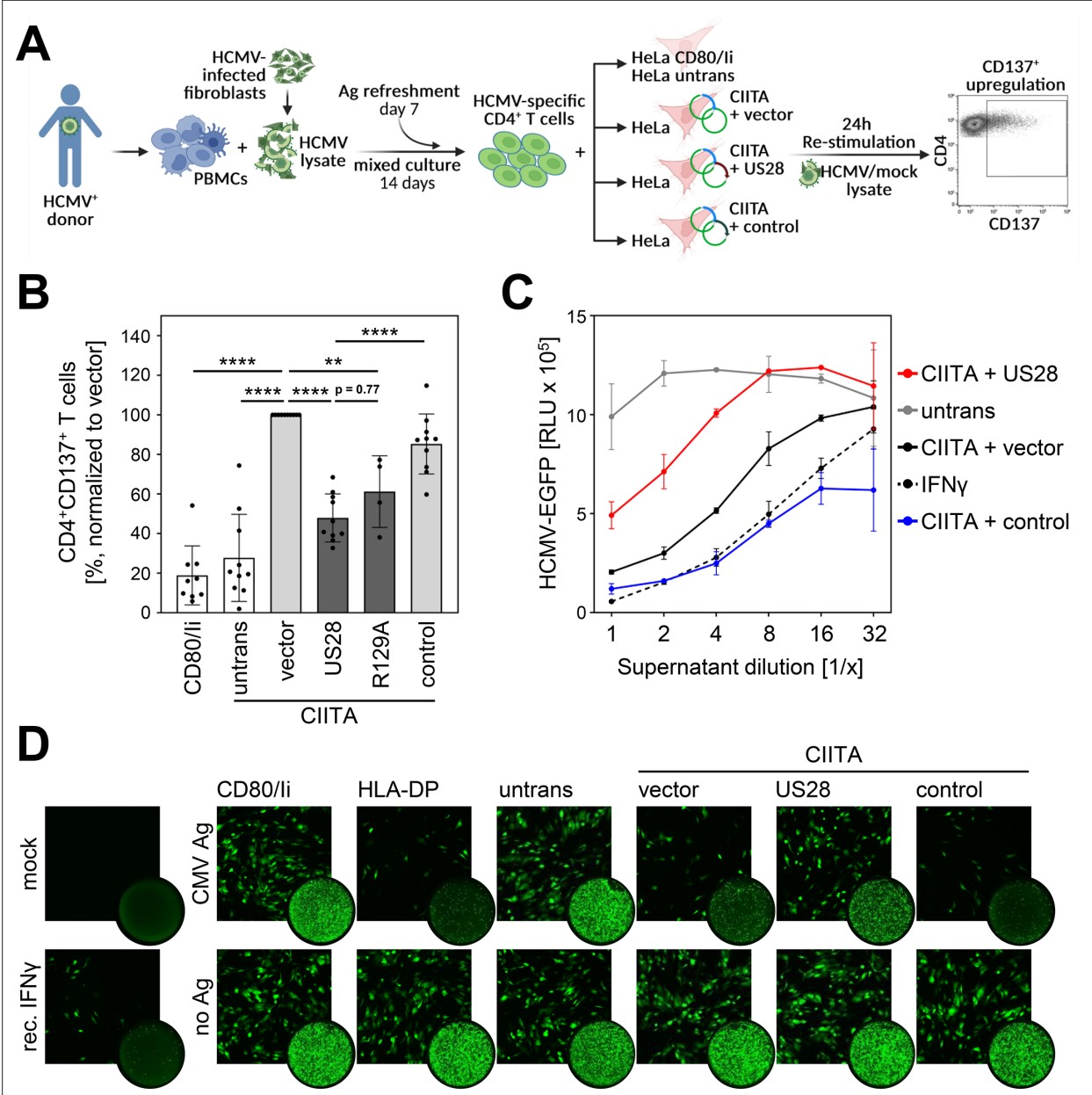

**Figure 8.** Activation of Human cytomegalovirus (HCMV)-specific CD4 + T cells is inhibited by pUS28. (**A**) Schematic overview of the experimental setup. Peripheral blood mononuclear cells (PBMCs) of Human cytomegalovirus (HCMV)-seropositive healthy donors were isolated, pulsed with HCMV lysate, and incubated for 14 d, with an antigen refreshment step at day 7. Afterwards, cells were co-cultured with HeLa cells that were either left untreated or were co-transfected with class II transactivator (CIITA) expression construct and pcDNA:US28-HA, pcDNA:US28-R129A-HA, pIRESNeo-FLAG/HA-EYFP, or empty vector, 48 hr prior to co-culture, and were re-stimulated with mock or HCMV lysate. HeLa cells only expressing CD80 and the invariant chain (HeLa CD80/Ii) served as further negative control. After 24 hr of incubation, the specific T cell response was quantified by flow cytometry as percentage of gated CD4 + T cells expressing the activation marker CD137. This panel was created using BioRender.com. (**B**) Activation of HCMV-specific CD4 + T cells was measured as described in (**A**). Proportion of CD137-positive T cells was normalized to T cells activated by HeLa cells transfected with CIITA expression construct and empty vector, and pulsed with HCMV lysate. Mean values ± SD are depicted (n=4–10 different donors). Significance was calculated by one-way ANOVA test. Comparisons are shown when statistically significant. (**C**) MRC-5 cells were incubated with supernatants from HCMV-specific CD4 + T cells (**B**) or recombinant IFNγ in serial dilutions for 24 hr. Next, cells were infected with BAC20-EGFP at an MOI of 0.05 and HCMV-induced EGFP expression was measured at 5 d post-infection (n=2). (**D**) MRC-5 cells were incubated and infected as in (**C**) and infected cells were visualized by fluorescence microscopy after 4 d of infection (square pictures) or whole-well imaging at 6 d post-infection (circle pictures). Untrans, untransfected. Vector, empty vector control. R129A, point mutation mutant of pUS28. Control, pIRESNeo-FLAG/HA-EYFP. Mock, uninfected. CMV Ag, HCMV lysate-treated. No Ag, mock lysate-treated.

The online version of this article includes the following figure supplement(s) for figure 8:

*Figure 8 continued on next page*

*Figure 8 continued*

**Figure supplement 1.** Activation of Human cytomegalovirus (HCMV)-specific CD4 + T cells is inhibited by pUS28.

Recent work with the MCMV model showed that the deletion of the *US28* homolog *M33* results in altered MHC-I presentation in an allotype-specific manner (H-2L$^d$ and H-2K$^d$ being downregulated in an *M33*-dependent manner, but not H-2D$^d$) (*White et al., 2022*). Our data show that pM33 targets mouse CIITA (*Figure 5E*), raising the intriguing question if pM33 also affects the MHC-II-dependent recognition of MCMV-infected cells by CD4 + T cells. Furthermore, it will be interesting to study if and how pM33 cooperates with the other MCMV-encoded GPCR pM78, which is necessary but not sufficient for a post-transcriptional attack on MHC-II molecules in the endosome (*Yunis et al., 2018*).

Previous studies focused on the interplay between HCMV infection and HLA-DR presentation, largely neglecting other HLA-II molecules such as HLA-DP. Work by *Miller et al., 1998* showed that

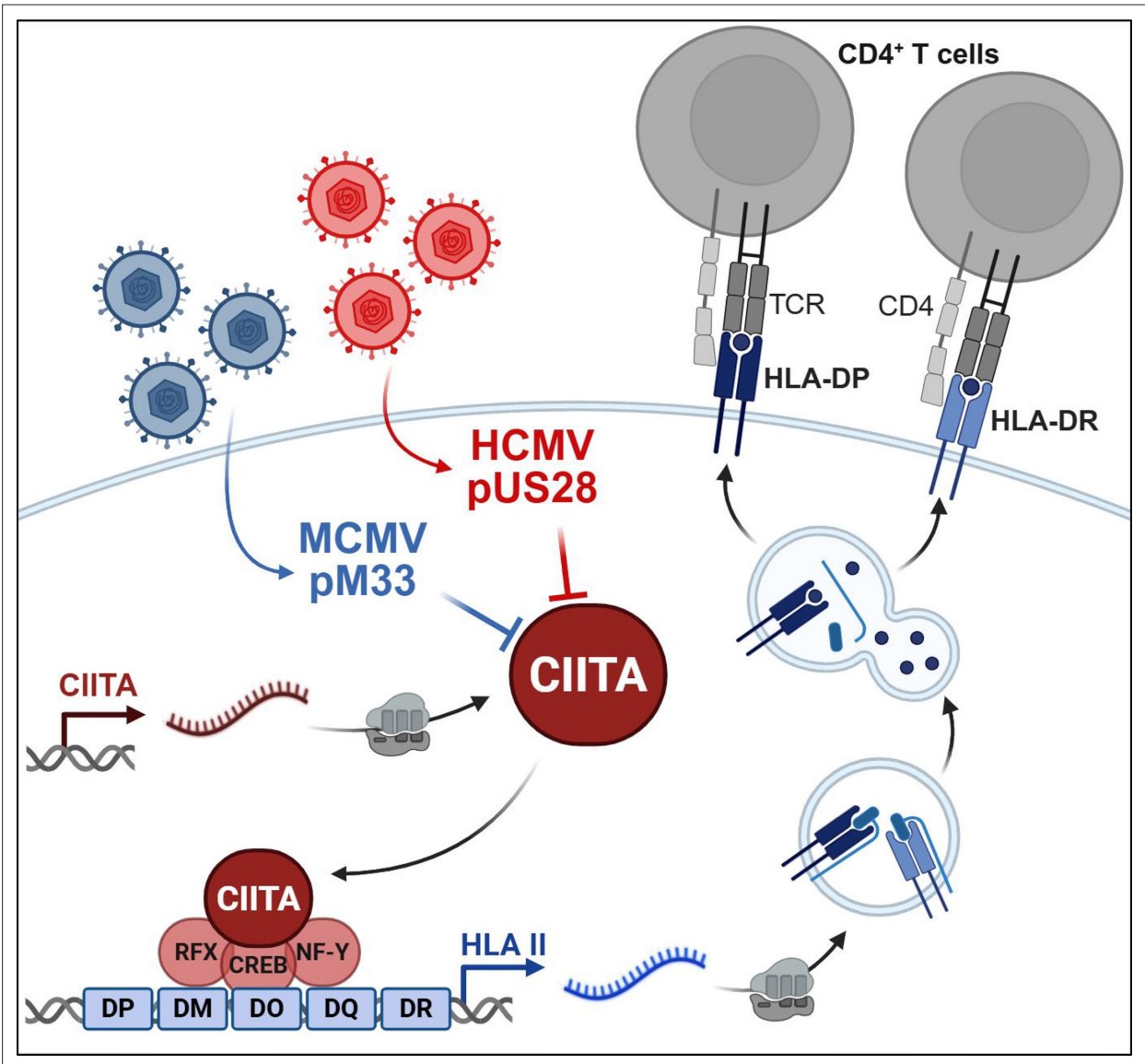

**Figure 9.** Model of the pUS28-mediated antagonism of class II transactivator (CIITA)-driven human leukocyte antigen (HLA)-II presentation and CD4 + T cell recognition. The findings shown in this publication are consistent with the following model: pUS28 acts as Human cytomegalovirus (HCMV)-encoded antagonist of CIITA and CIITA-driven HLA class II expression. The pUS28-dependent CIITA degradation is sufficient to decrease antigen presentation by HLA-II molecules including HLA-DR and HLA-DP, and to reduce the activation of HCMV-specific CD4 + T cells. The CIITA antagonism is evolutionarily conserved in the mouse cytomegalovirus (MCMV)-encoded pM33. This figure was created using BioRender.com.

HCMV counteracts IFNγ-induced gene expression, partly by inducing the proteasomal degradation of JAK1, leading to diminished CIITA and HLA-DRα induction when HCMV-infected cells are exposed to IFNγ (*Miller et al., 1998*). Furthermore, HCMV targets constitutive as well as induced CIITA-dependent HLA-II expression at multiple levels by a multipronged attack, comprising the inhibition of (I) IFN-JAK-STAT signaling (*Le et al., 2008*; *Le Trilling et al., 2020*; *Miller et al., 1998*), (II) constitutive CIITA transcription (*Lee et al., 2011*; *Sandhu and Buchkovich, 2020*), (III) CIITA protein stability (as shown here), and (IV) HLA-DR degradation and translocation (*Hegde and Johnson, 2003*; *Hegde et al., 2002*; *Tomazin et al., 1999*). This high level of redundancy may also explain why others concluded, based on loss-of-function experiments, that the *US* gene region comprising *US28* is dispensable for the inhibition of HLA-DR presentation (*Sandhu and Buchkovich, 2020*), while we observed a clear gain-of-function regarding HLA-II inhibition upon pUS28 expression and an impaired, but not completely abrogated, ability to decrease HLA class II expression by ΔUS28-HCMV.

HLA-II-restricted CD4 + T cell immunity to HCMV is crucial for the control of the lifelong infection by this virus. Its impairment in immunocompromised patients, including HCT recipients, is associated with considerable clinical risks. The present study is the first to identify a mechanism by which HCMV downregulates HLA-DP. HCMV accomplishes the CIITA degradation by pUS28 which occurs even in the absence of other HCMV-encoded proteins. Based on the aforementioned role of pUS28 for HCMV latency, it is tempting to speculate that pUS28 may shield latently infected cells from HLA-II-restricted CD4 + T lymphocytes.

In our introduction, we referred to the wealth of knowledge regarding the critical importance of CD4 + T cells for the immune control of cytomegaloviruses in mouse and rhesus models as well as in humans. These findings seem to contradict the multitude of CMV-encoded inhibitors of constitutive and induced HLA-II presentation. How can these two, at first glance mutually exclusive, facts be reconciled? The first argument is that cytomegaloviruses and their hosts are situated in an evolutionary red-queen race that establishes a hard-fought equilibrium. Thus, the residual CD4 + T cell-mediated immune control recognizing CMV-infected cells presenting diminished HLA-II levels may still be crucial for host survival despite the existence of viral inhibitors. Another intriguing possibility is that a relevant fraction of the HLA-II-mediated CD4 + T cell activation might be elicited by cells that are resistant to HCMV infections, such as the recently described HLA-DR +NKG2C+memory NK cells (*Costa-García et al., 2019*), which have been shown to be mediators of viral control in transplanted patients (*Davis et al., 2015*). The elucidation of the complex molecular mechanisms governing HCMV immune evasion in the immunocompetent and the immunocompromised host will provide important guidance for the design of tailored protocols of risk protection, e.g., by vaccination, targeted cellular therapies or drugs that interfere with pUS28-mediated CIITA degradation.

## Methods
### Cells and cell lines
HeLa cells (ATCC CCL-2), HeLa CD80/Ii (*Rutten et al., 2010*), HeLa TRex Flip-in cells (*Castello et al., 2012*) and BJ-5ta cells (ATCC CRL-4001), ARPE-19 cells (ATCC CRL-2302), and MRC-5 fibroblasts (ATCC CCL-171) were grown in Dulbecco modified Eagle medium (DMEM, Gibco) supplemented with 10% (v/v) FCS (Sigma-Aldrich), 100 µg/ml streptomycin/100 U/ml penicillin (Gibco), and 2 mM glutamine (Gibco) at 37 °C in 5% $CO_2$. Growth medium for BJ-5ta cells was further supplemented with hygromycin B (10 µg/ml, Invivogen), for HeLa TRex Flip-in cells with blasticidin (10 µg/ml, Invivogen) and normocin (100 µg/ml, Invivogen), and for HeLa TRex Flip-in:US28HA and HeLa TRex Flip-in:US27HA with geneticin (500 µg/ml, Invivogen), blasticidin (10 µg/ml, Invivogen), and normocin (100 µg/ml, Invivogen). UoC-B6 (CVCL_A304) cells were grown in RPMI 1640 medium (Gibco) supplemented with 10% (v/v) FCS (Sigma-Aldrich) and 100 µg/ml streptomycin/100 U/ml penicillin (Gibco) at 37 °C in 5% $CO_2$. PBMCs were obtained from healthy blood donors from the University Hospital Essen after informed consent under Ethical Review Board approvals 14–5961-BO and 16–6769-BO, in accordance with the Declaration of Helsinki. HLA typing was performed by next generation sequencing as described (*Lange et al., 2014*). All blood donors were HCMV seropositive and were selected according to their HLA-DPB1 and HLA-DRB1 typing matching the endogenously expressed HLA-DP and HLA-DR in HeLa cells.

## Generation of monocyte-derived dendritic cells

To obtain human dendritic cells, monocytes were isolated from PBMCs using a CD14 MicroBead kit (Miltenyi) according to the manufacturer's instructions, and purity was validated by flow cytometry. For maturation of monocytes into dendritic cells, monocytes were first cultured for 3 d in RPMI 1640 (c.c.pro) supplemented with 10% (v/v) heat-inactivated human serum (Merck), 50 U/ml IL-2 (Miltenyi), 10 ng/ml IL-7, 5 ng/ml IL-15 (R&D Systems), 10 ng/ml IL-4 (R&D Systems) and 100 ng/ml GMCSF (Miltenyi). Immature dendritic cells were further differentiated into mature dendritic cells by culturing for 48 hr in medium supplemented with 10 ng/ml IL-1β (Miltenyi), 10 ng/ml TNFα (Miltenyi), 1000 U/ml IL-6 (Miltenyi), and 1 mg/ml PGE2 (Sigma-Aldrich).

## Generation of stable cell lines

HeLa TRex Flip-in cells (*Castello et al., 2012*), kindly provided by Professor Matthias Hentze, were co-transfected with pOG44 (3 µg) and pcDNA5-FRT/TO:EF1prom-US28HA or pcDNA5-FRT/TO:EF1prom-US27HA (1 µg) using FuGENE HD transfection reagent (Promega). At 48 hr after transfection, cells were selected by culturing in DMEM (Gibco) supplemented with 10% (v/v) FCS (Sigma-Aldrich), 100 µg/ml streptomycin/100 U/ml penicillin (Gibco), 10 µg/ml blasticidin (Invivogen), 100 µg/

**Table 1.** Primer sequences.

| | Forward | Reverse |
|---|---|---|
| Cloning HCMV-US28HA | CGGCTAGCATGACACCGACGACGACGACC | CGCTCGAGTTAAGCGTAATCTGGAACCATCGTATGGGTACGGTATAATTTGTGAGACGCG |
| Cloning HCMV-US27HA | CGGCTAGCATGACCACCTCTACAAATAATC | CGCTCGAGTTAAGCGTAATCTGGAACCATCGTATGGGTACAACAGAAATTCCTCCTCCCC |
| Cloning HCMV-US29HA | CGGCTAGCATGCGGTGTTTCCGATGGTGG | CGGAATTCTTAAGCGTAATCTGGAACCATCGTATGGGTACTCGGAGGTGTCAACAACCC |
| QuikChange US28-R129A | CACGGAGATTGCACTCGATGCCTACTACGCTATTGTTTAC | GTAAACAATAGCGTAGTAGGCATCGAGTGCAATCTCCGTG |
| QuikChange US28-ΔC (Δ298–354) | CGCTCGAGTTAAGCGTAATCTGGAACATCGTATGGGTACTTGGTGCCCACGAAGACG | CGTCTTCGTGGGCACCAAGTACCCATACGATGTTCCAGATTACGCTTAACTCGAGCG |
| QuikChange US28-Δ314–354 | CGCTCGAGTTAAGCGTAATCTGGAACATCGTATGGGTAGAGTCGCTGGCGAAACTCG | CGAGTTTCGCCAGCGACTCTACCCATACGATGTTCCAGATTACGCTTAACTCGAGCG |
| QuikChange US28-Δ330–354 | CGCTCGAGTTAAGCGTAATCTGGAACATCGTATGGGTACCGACGCGAAAAGCTCATGC | GCATGAGCTTTTCGCGTCGGTACCCATACGATGTTCCAGATTACGCTTAACTCGAGCG |
| QuikChange US28-Δ346–354 | CGCTCGAGTTAAGCGTAATCTGGAACATCGTATGGGTACTCGTCGGACAGCGTGTCG | CGACACGCTGTCCGACGAGTACCCATACGATGTTCCAGATTACGCTTAACTCGAGCG |
| AD169-BAC2-UL131rep-Kana | TGCGCCGTGGTGCTGGGTCAGTGCCAGCGGGAAACCGCGGAAAAAACGATTATTACCGAAGGATGACGACGATAAGTAGGG | GCGTCCCAGTAATGCGGTACTCGGTAATAATCGTTTTTTTCCGCGGTTTCCCGCTGGCACCAACCAATTAACCAATTCTGATTAG |
| ΔUS28-Kana | CAGTCTCTCGGTGCGTGGACCAGACGGCGTCCATGCACCGAGGGCAGAACTGGTGCTATCCCAGTGAATTCGAGCTCGGTAC | CACGGGGAAAAGAGGGGCGGACACGGGGTTTGTATGAAAAGGCCGAGGTAGCGCTTTTTTGACCATGATTACGCCAAGCTCC |
| HLA-DPB1 (*Meurer et al., 2018*) | GCTTCCTGGAGAGATACATC | CAGCTCGTAGTTGTGTCTGC |
| HLA-DR (*Morimoto et al., 2004*) | GCCAACCTGGAAATCATGAC | AGGGCTGTTCGTGAGCACA |
| CIITA (*Sandhu and Buchkovich, 2020*) | AGCCTTTCAAAGCCAAGTCC | TTGTTCTCACTCAGCGCATC |

ml normocin (Invivogen), and 500 µg/ml geneticin (Invivogen) until cell clones were grown. Successful generation of cells with Doxycycline-inducible expression of US28HA or US27HA was validated by immunoblot analysis.

## Viruses, infection, and HCMV lysate generation

Virus stocks from the HCMV strain AD169 (*Le et al., 2008*), AD169-BAC2 (*Le et al., 2011*), AD169-BAC2 ΔUS2-11 (*Zimmermann et al., 2019*), and AD169-BAC20-EGFP (*Le Trilling et al., 2016*) were generated as previously described and propagated in MRC-5 cells (*Hengel et al., 1995*).

AD169-BAC2-UL131rep was generated by two-step red-mediated recombination of HCMV bacterial artificial chromosome (*Tischer et al., 2006*) using the primers AD169-UL131rep-Kana1 and AD169-UL131rep-Kana2 (*Table 1*) for PCR amplification and AD169-BAC2 as parental BAC. AD169-BAC2rep ΔUS28 was generated according to previously described procedures (*Tischer et al., 2006*; *Wagner et al., 2002*) using AD169-BAC2-UL131rep as parental BAC. Briefly, a PCR fragment was generated using the plasmid pSLFRTKn (*Atalay et al., 2002*) as template and primers ΔUS28-Kana1 and ΔUS28-Kana2 listed in *Table 1*. The PCR fragment containing a kanamycin resistance cassette was inserted into the AD169-BAC2-UL131rep by homologous recombination in *E. coli*, resulting in replacement of the US28 target sequence. Flp-mediated recombination was used to remove the kanamycin resistance cassette flanked by frt sites. Successful mutagenesis was confirmed by PCR analysis. Recombinant HCMV was reconstituted from HCMV BAC DNA by transfection with FuGENE HD transfection reagent (Promega) into permissive MRC-5 cells and further propagation of the virus in ARPE-19 cells.

Viral titers were determined by standard plaque titration on MRC-5 cells. All infections were conducted with centrifugal enhancement (900 g for 30 min).

For generation of HCMV lysates for T cell stimulation, BJ-5ta cells were infected with AD169-BAC2 at an MOI of 3 or mock-treated. At 4 d post-infection, cells were scraped, washed, and resuspended in PBS. After five freeze-thaw cycles, lysates were treated with ultra-sonication (two times 10 s) and centrifuged (1000 g, 20 min, 4 °C). Supernatants were used for T cell culture assays.

## Treatment with cytokines and inhibitors

Human IFNγ (PBL), IFNα (PBL), and TNFα (PeproTech) were used in the following concentrations: 200 U/ml, 200 U/ml, and 20 ng/ml, respectively.

Doxycycline (200 ng/µl, Sigma) was used to induce expression of the transgenes in HeLa TRex Flip-in:US28HA and HeLa TRex Flip-in:US27HA cells.

The following inhibitors were used to target different cellular degradation pathways: MG-132 (10 µM; Sigma-Aldrich), MLN4924 (2.5 µM; Active Biochem), TAS4464 (1 µM; MedChemExpress), Bortezomib (1 µM; Sellekchem), 3-Methyladenine (10 mM; Sigma-Aldrich), Bafilomycin (1 µM; Tocris), Chloroquine (50 µM; Sigma-Aldrich), Ammonium chloride (5 mM; Sigma-Aldrich), Z-VAD-FMK (50 µM; R&D), Pepstatin A (10 µM; Roth), E-64 (5 µM; Sigma-Aldrich), PMSF (1 mM; Roth), Protease Inhibitor Cocktail (1:200; Sigma-Aldrich), Dynasore (100 µM; MedChemExpress), or Decanoyl-RVKR-CMK (10 µM; Sigma-Aldrich).

## Protein stability determination by cycloheximide chase assay

Protein stability was determined in transfected HeLa cells (see transfection, plasmids, and mutagenesis) by treatment with 50 µg/ml cycloheximide (CHX, Roth). Cells were washed once in CHX-containing medium, followed by incubation in CHX-containing medium for indicated time periods. Finally, whole cell lysates were prepared and subjected to immunoblot analysis.

## Protein precipitation of cell culture supernatant

Cell culture supernatant of transiently transfected HeLa cells (see transfection, plasmids and mutagenesis) was collected. 400 µl supernatant were mixed with 1600 µl ice-cold acetone and incubated for 1 hr at –20 °C. After centrifugation for 90 min, 4 °C, and 13,000 g, the pellet was dried and resuspended in RIPA buffer. Subsequently, samples were subjected to immunoblot analysis.

## Transfection, plasmids, and mutagenesis

Transient transfection was performed using 1 or 2 µg plasmid DNA and 3.5 or 7 µl FuGENE HD transfection reagent (Promega) per 5×10⁵ cells. Cells were transfected with the following plasmids:

pUNO1-hCIITA (Invitrogen), pRP-humanCIITA-3xFLAG (VectorBuilder), pRP-3xFLAG-mouseCIITA (VectorBuilder), pcDNA3.1(+) (Invitrogen), pIRES$_{neo}$-FLAG/HA-EYFP (RRID:Addgene_10825, Gift from Thomas Tuschl *Meister et al., 2004*), pOG44 (Invitrogen), and HCMV ORF Expression Library (*Salsman et al., 2008*). Following plasmids were generated in this study: pcDNA3.1:US28HA, pcDNA3.1:US28HA-R129A, pcDNA3.1:US28HA-ΔC (Δ298–354), pIRES2-EGFP:US28HA, pIRES2-EGFP:US28HA-1–297, pIRES2-EGFP:US28HA-1–313, pIRES2-EGFP:US28HA-1–329, pIRES2-EGFP:US28HA-1–345, pcDNA3.1:US27HA, pcDNA3.1:US29HA, pcDNA5-FRT/TO:EF1prom:US28HA, and pcDNA5-FRT/TO:EF1prom:US27HA.

Cloning of pcDNA3.1:US28HA, pcDNA3.1:US27HA, and pcDNA3.1:US29HA was performed using the primers listed in *Table 1*. In order to generate US28 mutants, QuikChangeII XL Site-Directed Mutagenesis kit (Agilent Technologies) was used according to the manufacturer's instructions with pcDNA3.1-US28HA plasmid as the template DNA and respective primers (see *Table 1*). Cloning of pIRES2-EGFP and pcDNA5-FRT/TO:EF1prom constructs was performed by subcloning the respective gene sequences from the pcDNA3.1 constructs. All constructs were confirmed by DNA sequencing of the insert (LGC Genomics).

## Immunoblot analysis

For immunoblotting, whole cell lysates were prepared as described before in RIPA (*Trilling et al., 2009*) or 5 M urea buffer and equal amounts of protein were subjected to SDS polyacrylamide gel electrophoresis (SDS-PAGE). Proteins were subsequently transferred onto nitrocellulose membranes and immunoblot analysis was performed with the following antibodies: HA (Sigma-Aldrich, H6908), FLAG (M2, Sigma-Aldrich, F3165), β-tubulin (Cell Signaling, 2146), β-actin (Sigma-Aldrich, A2228), GAPDH (FL-335, Santa Cruz, sc-25778). Proteins were visualized using peroxidase-coupled secondary antibodies (rabbit-POD, Sigma-Aldrich, A6154; mouse-POD, Jackson ImmunoResearch, 115-035-062) and an enhanced chemiluminescence system (Cell Signaling Technology).

## Immunoprecipitation

Cells were lysed (150 mM NaCl, 10 mM KCl, 10 mM MgCl$_2$, 10% [v/v] glycerol, 20 mM HEPES [pH 7.4], 0.5% [v/v] NP-40, 0.1 mM phenylmethylsulfonyl fluoride [PMSF], 1 mM dithiothreitol [DTT], 10 µM pepstatin A, 5 µM leupeptin, 0.1 mM Na-vanadate, Complete protease inhibitor EDTA-free [Roche]). Lysates were centrifuged and immunoprecipitation (IP) antibody (anti-HA, HA-7, Sigma-Aldrich, H3663, or anti-FLAG, M2, Sigma-Aldrich, F3165) was added to the supernatant. Precipitation of immune complexes was performed with protein G-sepherose (GE Healthcare), benzonase (Sigma-Aldrich, E1014) digestion for 3 hr at 4 °C, and washing steps with 150, 250, and 500 mM NaCl-containing buffer. Samples were further processed by immunoblot analysis.

## Semi-quantitative RT-PCR

For semi-quantitative RT-PCR, total RNA was isolated from 1×10$^6$ cells using the RNeasy Mini kit (Qiagen) and digested with DNase I. Subsequent one-step RT-PCR (Qiagen) was performed using gene-specific primers listed in *Table 1*.

## Flow cytometry

For flow cytometry, cells were detached, washed with 2% FCS-PBS, and stained with labeled antibodies. For intracytoplasmic staining, cells were fixed in 4% PFA-PBS for 15 min at room temperature after detachment and washing steps. Permeabilization was performed with 1% saponin-PBS for 15 min at room temperature, followed by antibody staining. The following antibodies were used: HLA-DP-BV421 (B7/21, BD Biosciences, 750875), HLA-DP-APC (B7/21, Leinco Technologies, H240), HLA-DP-PE (B7/21, BD Biosciences, 566825), HLA-DR-PE (L243, BD Biosciences, 347401), HLA-DQ-PE (HLADQ1, BioLegend, 318106), HLA-DM-PE (MaP.DM1, BD Biosciences, 555983), CD137-APC (4B4-1, BD Biosciences, 561702), CD4-PE-Cy7 (SK3, BD Biosciences, 557852), CD8-Pacific Blue (B9.11, Beckman Coulter, B49182), CD3-Krome Orange (UCHT1, Beckman Coulter, B00068), CD57-FITC (TB03, Miltenyi Biotec, 130-122-935), CD14-FITC (MφP9, BD Biosciences, 345784), CD19-APC (HIB19, BD Biosciences, 555415), CD56-PE (N901, Beckman Coulter, A07788). Measurements were performed in a Gallios 10/3 cytometer (Beckman Coulter), using the Kaluza for Gallios Acquisition

software (Version 1.0, Beckman Coulter), or a BD FACSCanto II (BD Biosciences), using BD FACSDiva software (BD Biosciences). Data analysis was conducted with FlowJow (Version 10.8.1, Tree Star).

## T cell activation assay

To obtain HCMV-specific CD4 + T cells, PBMCs were pulsed with HCMV lysate (25 µg/ml) for 4 hr at 37 °C in 5% $CO_2$. Cells were cultured for 14 d in RPMI 1640 (ccpro) supplemented with 10% heat-inactivated human serum (Merck), 10 ng/ml IL-7, 1 ng/ml IL-12, and 5 ng/ml IL-15 (R&D Systems). Re-stimulation with irradiated (100 Gy), lysate-pulsed PBMCs was performed at day 7 of the co-culture. Afterwards, cells were cultured in RPMI 1640 (c.c.pro) supplemented with 10% heat-inactivated human serum (Merck), 50 U/ml IL-2 (Miltenyi), 10 ng/ml IL-7 and 5 ng/ml IL-15 (R&D Systems). After 14 d, expanded T cells were re-challenged for 24 hr with HeLa cells transfected with the respective plasmids and pulsed with mock or HCMV lysate (25 µg/ml). The specific T cell response was quantified by flow cytometry as a percentage of gated CD4 + T cells expressing the activation marker CD137 as previously described (*Meurer et al., 2018*).

## Antiviral activity of T cell supernatant

To evaluate the antiviral activity of supernatants from the T cell activation assay, MRC-5 cells were incubated with these supernatants in serial dilutions for 24 hr before cells were infected with HCMV-BAC20-EGFP at an MOI of 0.05. HCMV-induced EGFP expression was quantified using a Mithras² LB 943 Multimode Reader (Berthold Technologies, Software MikroWin 2010). Microscopy was conducted with a Leica DM IL LED Microscope (Software LAS V4.0) and a Bioreader-7000 Fz (BIO-SYS, Software EazyReader).

## LS-MS/MS sample preparation

Cell lysates were processed according to the SP3 protocol (*Hughes et al., 2019*) with minor modifications. Briefly, the cells were lysed in urea buffer (7 M urea, 2 M thiourea, 30 mM TRIS, 0.1% sodium deoxycholate, pH 8.5) and an aliquot of 10 µg was reduced with dithiothreitol (5 mM final concentration, 50 °C, 15 min), and alkylated using 2-iodoacetamide (15 mM final concentration, RT, 15 min). Subsequently, 100 µg of SP3-beads were added and the volume was adjusted to 100 µL using 50 mM ammonium bicarbonate (ambic). 170 µL of acetonitrile (ACN) were added and samples were incubated for 18 min. After washing the beads twice with 180 µL 70% EtOH and once with 180 µL ACN, 1 µg trypsin (SERVA Electrophoresis, Heidelberg, Germany) in 55 µL ambic was added and samples were digested overnight at 37 °C. Finally, the solution was transferred to a new vial, evaporated to dryness, and peptides were resuspended in 100 µL 0.1% trifluoracetic acid.

## Data-independent acquisition mass spectrometry

300 ng tryptic peptides per sample were analyzed in randomized order using a Vanquish Neo UHPLC coupled to an Orbitrap 480 mass spectrometer (both Thermo Scientific). The mobile phase A consisted of 0.1% formic acid (FA), mobile phase B of 80% ACN, and 0.1% FA. Peptides were loaded on a trap column (Acclaim PepMap 100, 100 µm × 2 cm, Thermo Scientific) using combined control with a loading volume of 20 µL, a maximum flow rate of 30 µL/min and a maximum pressure of 800 bar. Separation of peptides was achieved using a DNV PepMap Neo separation column (75 µm × 150 mm, Thermo Scientific) and a gradient from 1–40% B within 120 min and a flow rate of 400 nL/min at 60 °C. The MS parameters were set as follows: The RF lens amplitude was set to 55%, the MS1 scan range was 350–1450 m/z with a resolution of 120,000, a normalized AGC target of 300% and a maximum injection time of 54 ms. MS2 scans were acquired using a resolution of 30,000, a HCD collision energy of 30%, and a scan rage of 145–1450 with a normalized AGC target of 2500% and a maximum injection time of 80ms. A total of 40 isolation windows between 350 and 1450 m/z were cycled through with one MS1 scan being recorded after every 21 MS2 scans.

## Mass spectrometry data analysis

Protein identification and quantification was conducted using DIA-NN (v.1.8.1; *Demichev et al., 2020*) in library-free mode. The SwissProt database restricted to *Homo sapiens* as well as the Uniprot reference proteome for the Human cytomegalovirus (both ver. 2022_02) were used for peptide identification. Default settings were used, except for the neural network classifier, which was used in

double-pass mode, and protein inference, which was set to species-specific. The report file was filtered for all q-values≤0.01 using R (ver. 4.3.0; https://www.r-project.org/). Subsequently, protein quantities were calculated using the MaxLFQ algorithm as implemented in the DIA-NN R package. Missing data was imputed on protein level using the mixed imputation function from the imp4p package (*Gianetto et al., 2020*). Statistically significant differences between the experimental groups were assessed by means of ANOVA followed by Tukey's HSD post hoc tests. The ANOVA p-value was corrected for multiple testing according to the method of Benjamini-Hochberg. The significance threshold was set to pFDR ≤0.05, p posthoc ≤0.05 and a ratio of mean intensities ≥2 or ≤0.5.

## Quantification and statistical analysis

The resulting data were analyzed using GraphPad Prism software. The values are reported as Mean ± standard deviation (SD). Statistical significance was tested by applying the respective test indicated in the figure legends.

## Acknowledgements

We thank Matthias Hentze for kindly providing the HeLa TRex Flip-in cell line, Lejla Timmer, Kerstin von Ameln, and Sophie Eppler for excellent technical support, as well as the teams of the Fleischhauer and Trilling laboratories for insightful discussions. MT received funding from the Deutsche Forschungsgemeinschaft (DFG) through grants TR 1208/1–1 and TR 1208/2–1. KF received funding from the DFG through grant FL 843/1–1, the Deutsche José Carreras Leukämie Stiftung (DJCLS 20 R/2019), the Dr. Werner Jackstädt Stiftung, and the Joseph Senker Stiftung. We also acknowledge support by the Open Access Publication Fund of the University of Duisburg-Essen.

## Additional information

### Funding

| Funder | Grant reference number | Author |
| --- | --- | --- |
| Deutsche Forschungsgemeinschaft | TR 1208/1-1 | Mirko Trilling |
| Deutsche Forschungsgemeinschaft | TR 1208/2-1 | Mirko Trilling |
| Deutsche Forschungsgemeinschaft | FL 843/1-1 | Katharina Fleischhauer |
| Deutsche Jose Carreras Leukämie Stiftung | DJCLS 20R/2019 | Katharina Fleischhauer |
| Dr. Werner Jackstädt-Stiftung | | Katharina Fleischhauer |
| Joseph Senker Stiftung | | Katharina Fleischhauer |

The funders had no role in study design, data collection and interpretation, or the decision to submit the work for publication.

### Author contributions

Fabienne Maassen, Conceptualization, Data curation, Formal analysis, Validation, Investigation, Visualization, Methodology, Writing – original draft, Writing – review and editing; Vu Thuy Khanh Le-Trilling, Conceptualization, Resources, Data curation, Formal analysis, Supervision, Investigation, Visualization, Methodology, Writing – original draft, Writing – review and editing; Luisa Betke, Corinna Siegmund, Benjamin Katschinski, Antonia Belter, Investigation, Methodology, Writing – review and editing; Thilo Bracht, Resources, Data curation, Investigation, Methodology, Writing – review and editing; Malte Bayer, Investigation, Methodology; Tanja Becker, Barbara Sitek, Resources, Investigation, Methodology, Writing – review and editing; Denise Mennerich, Methodology, Writing – review and editing; Sebastian Voigt, Formal analysis, Writing – review and editing; Lori Frappier, Resources, Writing – review and editing; Katharina Fleischhauer, Conceptualization, Supervision, Funding acquisition,

Project administration, Writing – review and editing; Mirko Trilling, Conceptualization, Data curation, Supervision, Funding acquisition, Investigation, Visualization, Writing – original draft, Project administration, Writing – review and editing

### Author ORCIDs
Fabienne Maassen ⓘ https://orcid.org/0009-0004-3535-3727
Vu Thuy Khanh Le-Trilling ⓘ https://orcid.org/0000-0002-2733-3732
Luisa Betke ⓘ https://orcid.org/0009-0008-6767-6239
Corinna Siegmund ⓘ https://orcid.org/0000-0002-1371-0779
Malte Bayer ⓘ https://orcid.org/0000-0003-1060-8856
Benjamin Katschinski ⓘ https://orcid.org/0000-0002-1314-2820
Antonia Belter ⓘ https://orcid.org/0009-0005-4409-1447
Mirko Trilling ⓘ https://orcid.org/0000-0003-3659-3541

### Ethics
Human subjects: PBMCs were obtained from healthy blood donors from the University Hospital Essen after informed consent under Ethical Review Board approvals 14-5961-BO and 16-6769-BO, in accordance with the Declaration of Helsinki.

### Decision letter and Author response
Decision letter https://doi.org/10.7554/eLife.96414.sa1
Author response https://doi.org/10.7554/eLife.96414.sa2

---

## Additional files

### Supplementary files
Supplementary file 1. Global proteome analysis data.

MDAR checklist

### Data availability
All data generated or analysed during this study are included in the manuscript and supporting files; source data files have been provided.

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
