## [Editor Report]

This fundamental study substantially advances our understanding of viral immune evasion by identifying a novel player in this process and uncovering its mode of action. The evidence supporting the conclusions is compelling, with rigorous immunological assays and state-of-the-art biochemistry. The work will be of broad interest to virologists and immunologists.

---

## [Decision Letter]

[Editors' note: this paper was reviewed by Review Commons.]

---

## [Author Response]

General Statements [optional]

We really appreciate the very insightful and thorough evaluation of our study by all three reviewers. We are grateful to the reviewers for highlighting the relevance of our work and for their suggestions on how to improve the manuscript. We addressed all comments of the reviewers – in many cases through extensive new experiments and additional data.

Reviewer #1: In this study entitled "The human cytomegalovirus-encoded pUS28 antagonizes CD4^+^ T-cell recognition by targeting CIITA", Maassen et al. tried to identify viral genes/proteins downregulating the HLA class II molecule HLA-DP. In a transfection-based screening assay, they identified US28 as a viral gene downregulating CIITA-dependent HLA-DP expression in transfected HeLa cells. Other HLA class II molecules and other proteins expressed in a CIITA-dependent manner were downregulated as well. The authors went on to show that CIITA transcripts were not reduced in the presence of pUS28, but CIITA protein levels were massively reduced, suggesting that US28 downregulates CIITA on the post-transcriptional level. A signaling-deficient US28 mutant, but not a mutant lacking the cytoplasmic carboxy terminus, was capable of reducing CIITA levels. In a final set of T cell activation experiments, the authors showed that US28-expression in HeLa cells reduced HLA-II-dependent restimulation of CD4^+^ T cells.The data presented in the paper are generally very clean and convincing. However, not all conclusions are sufficiently supported by the data. A major weakness of the study is the fact that most experiments were done with transfected HeLa cells. Whether the proposed US28-mediated HLA-II downregulation occurs in HCMV-infected cells remains unclear. Moreover, the proposed pUS28-CIITA interaction was demonstrated in a single co-IP experiment and the mechanism of CIITA downregulation remains obscure. Hence, the conclusion that they have identified "a mechanism employed by HCMV to evade HLA-II-mediated recognition by CD4^+^ T cells" (abstract) is not justified.Reviewer #1: Major comments: 1. The mechanism of US28-dependent CIITA downregulation remains unresolved. The authors have made several attempts to clarify the mechanism, but these experiments have not met with success. Therefore, conclusions on the underlying mechanism should be toned down or removed.

We identified pUS28 as an HCMV-encoded CIITA antagonist. pUS28 acts post-transcriptionally and in the absence of other viral proteins by reducing the protein half-life of CIITA. In the revised manuscript, we further show that the capacity to downregulate CIITA is evolutionary conserved in the mouse cytomegalovirus (MCMV)-encoded G protein-coupled receptor (GPCR) pM33 (see new figure 5D), and that both pUS28 and pM33 target human and mouse CIITA (see new figure 5E). Conversely, the HCMV-encoded GPCR pUS27 does not affect CIITA. We also showed that the C-Terminus of pUS28 is indispensable for CIITA degradation and HLA-II downregulation (see new Figure 7C-I), but not for the physical interaction with CIITA (see new Figure 7H), suggesting that the C-terminus connects pUS28 with the degradation machinery while the aa 1-297 contains at least one relevant domain mediating the interaction with CIITA. A panel of inhibitors did not restore CIITA (see Figure 6 —figure supplement 4).

We feel that we made several important findings regarding the mechanism of HCMV-encoded HLA-II/CD74 antagonism. Indeed, the cellular co-factors recruited by the C-terminus of pUS28 mediating CIITA degradation remain to be identified. We clearly stated this point in the revised version of the manuscript and changed the last sentence of the abstract accordingly.

Reviewer #1: 2. The claim that US28-dependent CIITA downregulation occurs by pUS28 interacting with CIITA is based on a single co-IP experiment. The result would be more convincing if the authors could show the same interaction in a reverse IP, and ideally also in HCMV-infected cells. Is the US28 C-terminus required for this interaction?

We agree and performed the reverse experiment (see new Figure 6A), which showed that the pUS28/CIITA complex can be precipitated by both approaches. As stated above, the C-terminus is essential for CIITA degradation and HLA-II down-regulation (Figure 7C/F/G), but not for the interaction of pUS28 with CIITA (see new Figure 7H).

Reviewer #1: 3. The authors could not demonstrate US28-dependent HLA-II downregulation in HCMV-infected cells. Hence, they cannot conclude that HCMV employs this mechanism. Transfected HeLa cells are a somewhat artificial system. This does not invalidate the data, but one has to be careful when interpreting the data. Others have shown that HCMV downregulates CIITA transcript levels in myeloid cells (PMID 21458073 and 31915281). This apparent discrepancy could either be explained by several redundant mechanisms (as proposed by the authors of this manuscript) or by differences and limitations of the respective experimental systems.

Based on the studies mentioned by the reviewer, which documented a down-regulation of CIITA transcript levels, and the fact that our screen argues in favor of other antagonists of CIITA-induced HLA-DR and -DP induction (see Figure 2A-C; e.g., see the effect of pUL133 and maybe pUS19), we are convinced that redundancy indeed is an important aspect of HCMV-encoded CIITA/HLA-II downregulation. The reviewer is certainly aware of the fact that HCMV encodes an astonishingly broad arsenal of partially redundant and partially cooperative antagonists with regard to various other aspects of the immune system such as HLA class I presentation and NK-cell activation (see e.g., PMID: 25418469, 31666730). We discussed this redundancy on page 12 lines 342 and following in the manuscript.

To validate our findings in a system that does not rely on transient transfection, we generated doxycycline-inducible HeLa TRex Flip-in cell clones either expressing pUS28 or pUS27. Also in this model, pUS28 – but not pUS27- significantly down-regulated HLA-DP surface levels (see new Figure 5A).

Since we agree with the reviewer regarding the importance of a loss-of-function in the HCMV infection context, we generated wt and US28-deficient HCMV mutants by BACmid mutagenesis on a *UL131*/pentamer-repaired, *ULb*’-positive HCMV-AD169-BAC2 background (the non-repaired HCMV-AD169-BAC2 parental BACmid had been sequenced, annotated and described in PMID: 32075763 and its sequence is online available as MN900952) and compared them on cells that constitutively express HLA-II on their surface. In this setup, there is no need to induce the CIITA expression and HLA-II up-regulation by IFNγ, which is susceptible to HCMV-encoded IFN antagonists (please also see our responses to your THP-1 suggestion below). In this setting, we indeed observed an US28-dependent downregulation of HLA-DP (see new figure 5B) and HLA-DR (see new figure 5C) in the HCMV infection context.

Reviewer #1: 4. The possibility that US28 might downregulate CIITA in latently infected cells is intriguing. Have the authors tested this in an HCMV latency system, e.g. in infected THP-1 or Kasumi-3 cells? I acknowledge that such experiments are not trivial and may be beyond the scope of the present study. However, as latently infected cells express US28 but not the other viral genes previously shown to affect HLA-II expression (US2, US3, IE1 and 2), the latency model might a way to demonstrate biological significance in virus-infected cells.

Without an assessment of CIITA, others showed a pUS28-mediated downregulation of HLA-II mRNAs and proteins in THP-1 cells (PMID: 31796538). We cited this paper (page 11 lines 319 and following). Furthermore, Shnayder *et al.* showed a transcriptional downregulation of different HLA-II genes and CD74 in latently infected CD14+ monocytes, which we also cited in the revised manuscript (page 12 line 333). Our data provide an explanation for these important observations.

As suggested by the reviewer, we assessed THP-1 cells with regard to an HCMV-mediated downregulation of HLA-II. In our hands, THP-1 cells did not show a constitutive HLA-II expression. As expected, we were able to induce HLA-DP and HLA-DR in THP-1 cells by IFNγ. We also observed an inhibition of IFNγ-induced HLA-DP/-DR upregulation upon HCMV infection. However, this effect was not impaired when an US28-deficient HCMV was studied (see data in Author response image 1 which combines different attempts using different IFN treatment durations and infection times). We attribute this US28-independent inhibition of IFN-induced HLA-II upregulation to the HCMV-encoded IFN antagonists (discussed in the revised manuscript).

**Author response image 1. sa2fig1:** THP-I cells were either mock infected or were infected (MOI 3) with AD169-BAC2UL131rep (wt HCMV) or AD196-BAC-UL131repΔUS28 (HCMV ΔUS28). cells were either pre-treated with 200 IU/ml IFNγ for 24 — 48 h (different experiments) or were treated at 2 h post-infection. At 24 – 72 h post-infection (different experiments), cells were stained with anti-HLA-DP antibodies and analyzed by flow cytometry. The mean fluorescence intensity (fold MFI over background) values of HLA-II expression are depicted (n = 4-5).

THP-I cells were either mock infected or were infected (MOI 3) with AD169-BAC2UL131rep (wt HCMV) or AD196-BAC-UL131rePAUS28 (HCMV AUS28). cells were either pre-treated with 200 IJ/ml IFNγ for 24 — 48 h (different experiments) or were treated at 2 h post-infection. At 24 – 72 h post-infection (different experiments), cells were stained with anti-HLA-DP antibodies and analyzed by flow cytometry. The mean fluorescence intensity (fold MFI over background) values of HLA-II expression are depicted (n = 4-5).

The viral gene expression kinetics in Kasumi-3 are rather complex, including a phase of immediate early (IE) expression (PMID: 33731453) and according to the authors of the study “there was no preferential expression of UL138, US28, or RNA2.7 over the lytic gene UL32 when latency was established” (PMID: 30206173, 33731453). Therefore, we have refrained from applying Kasumi-3 cells as latency model.

As mentioned above, we used other cells (UoC-B6) that express HLA-II in the absence of IFNγ treatment. In this model, we observed an HCMV- and US28-dependent HLA-DP/-DR downregulation (see new figure 5B/C).

Reviewer #1 (Significance (Required)): Recognition of virus-infected cells by CD4^+^ T cells is an important immune defense mechanism. Viruses like HCMV have evolved numerous immune evasion mechanisms. Previous studies have identified HCMV proteins targeting HLA-II for degradation (US2, US3) or downregulating CIITA transcription (probably IE1+IE2). The findings of the present manuscript now demonstrate that US28 is capable of contributing to HLA-II downregulation. This is potentially of great significance as US28 is expressed in latently infected cells. However, the significance of the present study is limited by the fact that the studies were done in transfected HeLa cells, not in HCMV-infected cells, and that the mechanism of CIITA post-transcriptional downregulation remains unknown. In its present form, the study should be of interest for virologists and immunologists interested in new viral immune evasion strategies. The significance and appeal to a wider audience would be massively increased if the authors could clarify the mechanism or show its importance in virus-infected cells (or both).

As mentioned above, we confirmed the effect in another cell model (stable doxycycline-inducible cell clones [see new figure 5A]) and extended the effect to HCMV-infected cells (see new figure 5B/C). Furthermore, we show that the effect is evolutionary conserved in the MCMV homolog pM33 (see new figure 5D/E), allowing future studies in the MCMV mouse model.

Reviewer #2 (Evidence, reproducibility and clarity (Required)): In this study, Maaßen and colleagues investigate how HCMV US28 functions to antagonize the class II Transactivator (CIITA) transcription factor and subsequent HLA class II expression. Through physical interaction with CIITA, US28 triggered a post-transcriptional decline in CIITA protein, leading to reduced cell surface expression of HLA class II molecules, including HLA-DR, HLA-DQ, HLA-DM, CD74, and HLA-DP. Moreover, the authors show that US28-mediated degradation of CIITA hindered the activation of HCMV-specific CD4^+^ T cells. Strengths of this article include the rigorous methodologies and analysis, clear and concise writing, and relevance within a clinical context. Despite the strengths, a few deficiencies were identified.Reviewer #2: There is a lack of information regarding how the expression library screen was performed and how hits were determined/chosen for further analysis.

In the revised manuscript, we elaborated on how the screen was performed and why we focused on pUS28 for further analyses.

Reviewer #2: Figure 1: Lacks statistical analysis and the number of replicate experiments that were performed. Many figure legends lack the information regarding the number of replicate experiments that were performed.

We added this information for Figure 1 (see new Figure 1E) and throughout the manuscript.

Reviewer #2: While the observation that US28 antagonizes CIITA is well supported, the mechanism behind the antagonism is somewhat lacking. These findings have major implications for understanding the immune response to HCMV, particularly in immunocompromised patients where impaired HLA-II presentation poses clinical risks. The authors suggest that a comprehensive understanding of the molecular mechanisms governing HCMV immune evasion could guide the development of tailored protocols for risk protection, such as vaccination, cellular therapies, or drugs targeting US28-mediated CIITA degradation.

Although we made additional observations regarding the molecular determinants of CIITA binding and degradation (see new Figure 7) and tested a panel of degradation inhibitors (see Figure 6 —figure supplement 4), the cellular co-factors of this degradation remain to be identified in future studies.

Reviewer #2: Figure 2: Lacks information regarding how hits from the expression screen were determined. It would be helpful to understand the selection criteria.

Our criteria were a relevant inhibition of the CIITA-driven HLA-DP upregulation (assessed in the screen shown in Figure 2A), novelty (see revised text), and reproducibility (Figure 2B and data not shown). Our focus on pUS28 for further experiments is explained in the revised manuscript. A potential biological relevance of the minor effect on HLA-DP upon co-transfection of CIITA and pUS19 is currently under investigation – but beyond the scope of this manuscript.

Reviewer #2: Figure 3: While the blots here support the authors claim that US28 antagonize CIITA at the post transcriptional level, it would be beneficial to make these observations within the context of viral infection.

As described in more detail in our response to Reviewer #1, we indeed confirmed the US28-dependent downregulation of HLA-DP and HLA-DR in the HCMV infection context (see new Figure 5B/C).

Reviewer #2: Figure 5B: Labeling of this panel is confusing. The authors should attempt to relabel, or perform the experiment again and run samples on one gel.

All samples were run on a single gel, blotted on one membrane, and assessed in one immunoblot. We present the results in the cropped and rearranged manner, to enable readers to directly compare corresponding time points. We have now included the uncropped immunoblot as Supplementary Figure in the revised manuscript (see new Figure 6 —figure supplement 3). We also optimized the labelling (see newly labelled Figure 6B).

Reviewer #2: Figure 5: data for the claim that US28-mediated antagonism of CIITA is independent of neddylation, proteasomal degradation, etc. should be shown if the authors wish to make this claim.

As suggested by the reviewer, we added the data of these experiments to the revised manuscript (see new Figure 6 —figure supplement 4).

Reviewer #3 (Evidence, reproducibility and clarity (Required)): Maaßen and colleagues followed up on their observation that infection with HCMV mutants lacking pUS2 and 3 impaired HLA-DP expression of IFN-γ treated MRC-5 cells. This phenomenon is interesting because pUS2 and 3 are viral components that have been shown earlier to degrade HLA-DR α and HLA-DM α and thus inhibit the formation of HLA-DR α/β heterodimers. These data indicated that in addition to pUS2 and 3 some other MHC II inhibitor must be encoded by HCMV. The authors tested this hypothesis by analyzing a HCMV gene expression library for the presence of a new HLA-DP antagonist. Indeed, the data revealed pUS28 as a new MHC II inhibitor that exhibited a posttranscriptional effect on CIITA, which is the key regulator of MHC II expression. In in vitro stimulation experiments, pUS28 impaired activation of antigen-specific CD4^+^ T cells.The study provides new and important information on how HCMV evades human immunity. The shown data support the main conclusions. Nevertheless, inclusion of some additional controls would facilitate understanding the overall concept.Reviewer #3: Minor points: A key element of this study is that IFN-γ induces MHC II expression on MRC-5 fibroblasts. Nevertheless, professional antigen presenting cells such as dendritic cells express MHC II independent of any stimulation. Therefore, in Figure 1 in addition to IFN-γ induced DP expression on MRC-5 fibroblasts, DP and DR expression of dendritic cells should be shown. This could be easily done by analysis of dendritic cells in PBMC. Furthermore, the authors should show DP and DR expression of dendritic cells with and without IFN-γ stimulation. Most likely, DP and DR expression of untreated dendritic cells is significantly higher than DP expression of IFN-γ treated MRC-5 cells. It is important to include such controls to avoid the impression that IFN-γ treated fibroblasts can have similar functions as professional antigen presenting cells.

We added the comparison (see new Figure 1 —figure supplement 1). As the reviewer anticipated, the HLA-II expression of MoDCs (even in the absence of IFNγ) exceeds the expression reached by IFNγ-exposed fibroblasts.

However, there is clear evidence that HLA/MHC-II expression occurs in cells other than DCs such as human vascular endothelial and dermal fibroblasts (PMID: 6415484, 30421030) and that the induced HLA-II expression is immunologically highly relevant. For example, a loss of an MHC-II gene (H2-Ab1) in intestinal epithelial cells (IEC) resulted in less severe DSS- and T-cell transfer-induced colitis, and 50% of mice with an IEC-specific MHC-II loss died after infection with *Citrobacter rodentium*, whereas none of the control mice died (PMID: 32589883). An IEC-specific deletion of MHC-II can also prevent the initiation of lethal GvHD in the GI tract (PMID: 31542340). Furthermore, monocytes and macrophages express HLA-II and can be infected by HCMV. Senescent fibroblasts also seem to express functionally relevant levels of HLA-II (PMID: 37001502). Thus, we are convinced that HCMV “has good reasons” to target the CIITA-driven inducible HLA-II expression and the recognition by IFNγ-producing CD4^+^ T cells (Figure 9). Our observation that the MCMV-encoded GPCR pM33 also targets CIITA (see new Figure 5D/E) allows such questions to be addressed experimentally in model systems.

Reviewer #3: The conclusion in the last paragraph of the discussion that NKG2C+ memory NK cells might have activated antigen-specific CD4^+^ T cells is confusing. In the end, professional antigen presenting cells that have taken up viral proteins most probably stimulated antigen-specific CD4^+^ T cells. And since antigen presentation on MHC II is independent of infection of the antigen presenting cell, it is difficult to understand why under such conditions a red-queen race should have been taken place.

The relevance of professional antigen-presenting cells (pAPC), which remain uninfected but take up exogenous antigens and present them, is indisputable. However, we are convinced that (I) HCMV does effect cells that do express HLA-II (please see our response to the previous question) and (II) that the well-described inducibility of CIITA and the HLA-II locus by IFNγ is immunologically relevant. In accordance with this importance, we found that ability to target CIITA is evolutionary conserved in MCMV and HCMV. We favour the hypothesis that the selective pressure that led to the evolution of CMV-encoded GPCRs targeting CIITA was elicited either by HCMV-specific CD4^+^ T cells either cytotoxic ones (PMID: 37965314, 28392788, 27606804, 17158960) or bystander T cells that upon activation produce antiviral cytokines such as IFNγ. By comparing wt-MCMV with ΔM33-MCMV infections, we will be able to test such hypotheses in the future.

We rephrased the sentence regarding the NK cells in the revised manuscript.

Description of the revisions that have already been incorporated in the transferred manuscriptReviewer #1 6. Figure 2. The use of "empty" for mock-transfected cells is confusing, particularly as empty vector-transfected cells are labeled "vector".

We changed the description in the transferred manuscript.

Reviewer #2: Many figures lack statistical analysis to support the claims made by the authors. Where applicable, the authors should include these or explicitly state that only significant comparisons are shown.

We added this to the transferred manuscript.

Reviewer #2: Minor Comments: Labeling for immunoblots should be clearer regarding transfection conditions. The terms "empty" and "vector" is ambiguous and is not descriptive enough for the conclusions the authors are drawing.

We changed the description in the transferred manuscript.

Reviewer #2: Many figures, lack statistical analysis supporting the claims being made. If observations do not reach statistical significance, these should be explicitly stated within the legends. (i.e. comparisons are shown where statically significant).

We added this information in the transferred manuscript.

Reviewer #2: The authors should move away from making conclusions without showing any data substantiating their claims.

We added to the transferred manuscript the new Supplementary Figures that contain data previously referred to as “data not shown”.

Reviewer #2: Minor grammatical and citation errors were identified throughout the manuscript. The authors should carefully read and fix any errors prior to publication.

We did our best to fix the typos.

Reviewer #3: In Figure 6b the proportion of CD137-positive T cells normalized to T cells activated by HeLa cells transfected with CIITA and pulsed with HCMV lysate is shown. In this kind of data presentation, the magnitude of the original effect, i.e., the percentage of CD4^+^ T cells that after 24 h of stimulation is CD137-positive, remains unclear. Therefore, it is recommended to first show actual data and then relative values.

We added the actual data as Supplementary Figure to the transferred manuscript.

Reviewer #3: In the figure legend it is stated n = 4-10. Does this mean that T cells from different donors of the corresponding numbers have been tested? Or have T cells from some donors been tested more than once? More precise information should be given here.

4-10 different donors were assessed. We added this information in the transferred manuscript.

Reviewer #1: Minor comments: 5. Figure 2. Why was US29 used as a control and not US27 as in other experiments. The authors pointed out themselves that US27 is probably an ideal control for US28 as both genes encode related GPCRs.

The experiment shown is one of the earlier experiments. At that time, we simply had not yet cloned an epitope tagged version of pUS27. The reviewer is right that pUS27 is more closely related to pUS28 than pUS29. However, the comparisons with both pUS29 and pUS27 in different experiments show that the effect elicited by pUS28 is special. We compared pUS28 with pUS27 in various other figures and there we prefer to keep the figure as is.

Reviewer #2: Figure 3: While semi-quantitative RT-PCR is a useful method for determining the levels of mRNA, it would be beneficial to conduct RT-qPCR experiments to support the claim that US28 does not affect CIITA mRNA levels.

We appreciate the suggestion. In our hands, qPCRs using previously described CIITA primers (used in PMID: 31915281) frequently produce a combination of a specific and an unspecific amplificate. We applied semi-quantitative RT-PCRs and separated the bands in a gel rather than performing RT-qPCRs in order to exclude that unspecific amplificates compromise our result.

Reviewer #3: Considering the higher MHC II levels expressed by dendritic cells, it would be interesting to see to which extent pUS28 expression reduces MHC II expression of dendritic cells. Such experiments can be performed by lentiviral pUS28 expression for example in monocyte-derived dendritic cells.

From a mechanistic point of view, this is an intriguing experimental suggestion. Obviously, HCMV-encoded antagonists can be applied as “molecular scissors” to study the pathways that they antagonize. We hope that pUS28 – like the HLA-I inhibitors in the US gene region – will also be used by others in this way. As the reviewer argued herself/himself, most of these cells will not be infected by HCMV. Therefore, we think that such experiments are simply beyond the scope of the first description of the effect.